# On the effectiveness of communication strategies as non-pharmaceutical interventions to tackle epidemics

Alejandro Bernardin[1,2☺], Alejandro J. Martínez[1,3☺]*, Tomas Perez-Acle[1,2,3]*

**1** Computational Biology Lab (DLab), Fundación Ciencia & Vida, Santiago, Chile, **2** Centro Interdisciplinario de Neurociencia de Valparaíso, Facultad de Ciencias, Universidad de Valparaíso, Valparaíso, Chile, **3** Facultad de Ingeniería y Tecnología, Universidad San Sebastián, Santiago, Chile

☺ These authors contributed equally to this work.
* martinez@dlab.cl (AJM); tomas@dlab.cl (TPA)

**Data Availability Statement:** Our data is available for open and free access, with doi: 10.5281/zenodo.5518221 https://zenodo.org/record/5518221#.YUk68LpKhhE.

## Abstract

When pharmaceutical interventions are unavailable to deal with an epidemic outbreak, adequate management of communication strategies can be key to reduce the contagion risks. On the one hand, accessibility to trustworthy and timely information, whilst on the other, the adoption of preventive behaviors may be both crucial. However, despite the abundance of communication strategies, their effectiveness has been scarcely evaluated or merely circumscribed to the scrutiny of public affairs. To study the influence of communication strategies on the spreading dynamics of an infectious disease, we implemented a susceptible-exposed-infected-removed-dead (SEIRD) epidemiological model, using an agent-based approach. Agents in our systems can obtain information modulating their behavior from two sources: (i) through the local interaction with other neighboring agents and, (ii) from a central entity delivering information with a certain periodicity. In doing so, we highlight how global information delivered from a central entity can reduce the impact of an infectious disease and how informing even a small fraction of the population has a remarkable impact, when compared to not informing the population at all. Moreover, having a scheme of delivering daily messages makes a stark difference on the reduction of cases, compared to the other evaluated strategies, denoting that daily delivery of information produces the largest decrease in the number of cases. Furthermore, when the information spreading relies only on local interactions between agents, and no central entity takes actions along the dynamics, then the epidemic spreading is virtually independent of the initial amount of informed agents. On top of that, we found that local communication plays an important role in an intermediate regime where information coming from a central entity is scarce. As a whole, our results highlight the importance of proper communication strategies, both accurate and daily, to tackle epidemic outbreaks.

**Funding:** This work was partially supported by the Programa de Apoyo a Centros con Financiamiento Basal AFB 170004 to Fundación Ciencia & Vida (www.cienciavida.org) and the Instituto Milenio Centro Interdisciplinario de Neurociencia de Valparaíso ICM-ECONOMIA P09-022-F (cinv.uv.cl). This material is based upon work supported by the Air Force Office of Scientific Research under award number FA9550-20-1-0196. AB acknowledges FIB-UV scholarship from Universidad de Valparaíso (www.uv.cl). AJM acknowledges support from the Agencia Nacional de Investigación y Desarrollo de Chile (ANID) under Grant No. 3190906 (www.anid.cl). The authors also acknowledge the National Laboratory for High Performance Computing (NLHPC), Universidad de Chile. Powered@NLHPC. The funders had no role in study design, data collection and analysis, decision to publish, or preparation of the manuscript.

**Competing interests:** The authors have declared that no competing interests exist.

# Introduction

The spread of infectious diseases is nowadays an important health issue worldwide, killing about 8.5 million people yearly [1]. More recently, the broad and diverse impact of the COVID-19 pandemic around the world has demonstrated that human behavior [2–4] and communication [5–7] are both key components in the propagation, control [8, 9], and mitigation of epidemics [10], specially in the absence of pharmaceutical interventions. Despite this certainty, the actual relationship between communication and human behavior still remains unclear. For instance, questions such as *Which would be the reaction of a certain population during the spread of an epidemic disease?* seems to depend strongly on both sociological and communication factors [11–13], among others, making the problem extremely complex to tackle. Therefore, it appears to be essential to integrate socio-cultural factors along with epidemiological models, in order to accurately describe the temporal evolution of an epidemic.

Previous contributions focused on how media [14–23] and information [24–27] affects the spread of infectious diseases, (for detailed systematic reviews please refer to [28–30]). Main conclusions from this research area are summarized as follow: (i) human response depends on the specific disease being dispersed together with social, cultural, political, and economic factors characterizing the population in which the disease spread; (ii) appropriate data is required to identify human behaviours that are key to regulate the spread of the disease; (iii) agent models are suitable tools to study the effect of behavioral changes on a population under an epidemic situation; and (iv), social media and massive data strategies should be both considered at the moment of fitting and feeding models with real data [28]. Furthermore, an interesting emerging concept is the existence of a "Behavioral Immune System" [31], describing how the adoption of preventive behaviors could help people to reduce their probability of resulting infected during an epidemic. Notably, while vaccine coverage to deal with the COVID-19 pandemic remains low, particularly for developing countries, the existence of an actual Behavioral Immune System in the population is one of the best protective front lines we can rely on.

An interesting approach relating communication with the spreading of an infectious disease was proposed by Funk et al. in 2009 [32], by introducing the concept of "*awareness*" as a mathematical association between agents's proximity and their susceptibility to the infection. In doing so, the authors formulated a mathematical model describing how awareness spreads in population, coupled to an SEIR epidemiological model. Funk's model assumes that: (i) awareness is a positive-definite function of time and depends on how information's quality is distributed among susceptible individuals at a given time; (ii) information's quality decays in time unless agents get exposed to new sources of information of higher quality and also decays when it is transferred from one person to another; (iii) information sources can be of local or global nature, and they can conduce to a variety of different preventive measures adopted by agents. However, the epidemiological consequence in the system is the same in all cases: a reduction in the infection rate, independent from any information strategy. Using this scheme, the authors argued that self-initiated reactions made by individuals under the influence of a certain degree of awareness can be crucial to the epidemic fate. Interestingly, they observe that a disease can be completely stopped from spreading, only if the population awareness diminish the basic reproductive number of disease, $R_0$, below a certain threshold.

In this work, we push forward these ideas to investigate how different communication strategies, extending from local to global ones, may produce a quantifiable effect in the outcome of the spreading of an infectious disease. Throughout an exhaustive numerical analysis, we compare populations having different characteristics when adopting preventive measures based on the situational awareness. To do so, we relied on an agent-based model (ABM) framework in which the infection propagates through a host population, from agent to agent, following a

probabilistic process that depends on both the agents' proximity and their situational aware-
ness about the disease. Furthermore, agents can acquired information along time either from
an agent-to-agent interaction or from a central entity, reducing temporally their disease sus-
ceptibility, following the same logic as in the model proposed by Funk et al.

Our results suggest that informing a small fraction of the population has a remarkable
impact on the spread of the disease, compared to a situation where the population is not
informed at all. Moreover, communication strategies relying on a daily basis largely outper-
form the delivery of promptly information, delivered as soon as the disease spreads. Of note,
our models also suggest that the initial number of informed agents is irrelevant to the outcome
of an epidemic when information is not replenished over time. On top of that, communication
between agents plays a crucial role, for some societies, when information is scarcely replen-
ished, becoming irrelevant when large amount of information is available on a periodic basis,
from a central entity.

## Materials and methods

This section is divided in two parts. First, we introduce and contextualize our model, without
mathematical nor computational technicalities, instead we provide a discussion focused on the
learning that the 2014–2016 Ebola outbreak in West Africa, left us. This discussion is carried
out with special emphasis in the interplay between the propagation of a disease and communi-
cation factors in a society. Then, we go through the details of our model, discussing its applica-
bility and limitations.

### An SEIRD epidemiological model with communication

Among all possible infectious diseases, we decided to use the Ebola virus disease (EVD) in this
study for several reasons. Even though we are in the middle of a COVID-19 pandemic, and
considering the worldwide context it would have made more sense to work in those lines, our
understanding about EVD is fare more mature than that of COVID-19. Moreover, a vast liter-
ature regarding the dynamics of EVD has been published since 2014 [33–40], arriving to a cer-
tain degree of consensus about fundamental characteristics of this disease such as its
spreading, lethality and mortality, basic and time-dependant reproductive numbers, $R_0$ and $R_t$,
respectively, having an $R_0$ for Ebola virus within the interval 1.51–2.53 [33, 37, 40]. The latter
is extremely relevant for us, as our intention is to explore cases where, due to the action of
exchanging information, the result is a feasible reduction of the disease's negative impact in
the population. In other words, this can be interpreted as the possibility of modulating $R_t$ by
means of a strategy of information delivery, reaching a $R_t < 1$ at some point along the
simulation.

Furthermore, the extreme symptoms of EVD and the sociological impact of its outbreaks
can lead to undesired social behaviors from both authorities and the population such as fear,
discrimination, overreaction in the implementation of public policies, and the rejection of sci-
entific evidence. Examples of these anomalous behaviors are extensively discussed in [34, 41–
44], showing why considering human behavior, and not only epidemiological factors, is essen-
tial to understand the dynamics of the spreading of infectious diseases. Of note, previous EVD
outbreaks were successfully controlled by implementing public policies that helped to prevent
contagion [45–49]. Thankfully, the situation can potentially change now, as there are two EVD
vaccines that were approved between 2019 and 2020 for the Zaire strain: rVSV-ZEBOV
approved by the FDA [50, 51] and ZEBOV/MVA-BN-Filo which was approved by the Euro-
pean Union [52, 53]. However, considering the case of the current COVID-19 pandemic, and
moreover future epidemics, producing models to study how non-pharmaceutical

interventions based on communication strategies may impact on the spreading of infectious diseases, is crucial.

When it comes to modelling the dynamics of an epidemic, the compartmental models, first proposed by Kermack and McKendrick in 1927 [54], have shown their success and flexibility. Under this scheme, individuals are assigned to specific compartments depending on their current epidemiological state, and they can transit from one compartment to another along time, depending on the specific disease and dynamics. Common compartments are *Susceptible* (S), *Infected* (I), and *Removed* (R), among others. Using these elements as building blocks we may assemble a variety of epidemiological models such as the SIS (susceptible-infected-susceptible) or the SIR (susceptible-infected-removed) models. The former has been used to describe diseases such as the common cold or influenza [55, 56], whilst the latter to describe, for instance, measles [57, 58]. When modeling the EVD dynamics, the SEIRD model has shown to have good agreement with real data [36, 40]. Here, the E and D compartments stand for *Exposed* and *Dead*, respectively. Including these compartments explicitly defines an incubation period —i.e. defined as exposed state E—and, unlike other diseases, that susceptible individuals can still get infected with EVD through the contact with dead individuals, if no proper precautions are taken into consideration. Notably, despite controversy, this seems to be also the case of the COVID-19 pandemic [59–62].

Finally, to implement a model in which we may study the influence of communication strategies on the spread of EVD, we followed the approach proposed by Funk et al. [32], assuming that only trustworthy information is present in the system. Despite that the existence and importance of unreliable information, particularly in the form of *fake-news*, has been widely discussed [63, 64], we decided to focus only in reliable information as a first approach. Thus, the influence of fake-news during the spread of epidemic diseases, will be explored elsewhere. As a consequence, considering only truthful information implies that individuals having information are less likely to get infected than that of the ones without information. In our model, information can spread by three mechanisms: i) we consider that individuals can acquire information from a central entity corresponding to a global source of information; ii) through individual-individual interaction, which is a local source; and iii) when agents get infected and hence, they become informed. Of note, under this approach, individuals can affect each other not only from the epidemiological point of view –i.e. transmitting the disease–, but also because they disseminate information in the system. In the following section, we extend these ideas into a more technical and mathematical description of our model and simulations.

## Details of our ABM

We implemented our model using an ABM scheme having 10,000 agents in Netlogo 6.1.1 [65]. Netlogo is a free software designed to run multi-agent simulations, with a large set of included functionalities, having an extensive user community. It supports agent dynamics embedded in space, which makes it ideal for simulations in epidemiology when the dependency for the spreading dynamics on the spatial proximity between agents, is a desired characteristic [66–68].

In our simulations, an agent is characterized by the parameters $z_i$, $Q_i$, and $r_i$, where $z_i$ denotes the agent's position in space, $Q_i$ denotes its epidemiological state, and $r_i$ denotes its information state. The suffix $i$ denotes the $i$th agent. For simplicity, we assumed that each agent moves in a 2-torus (a 2D space with periodic boundary conditions) following a 2-D random walk (with unitary steps following random directions) and without affecting or being affected by other agents, at least in terms of its spatial dynamics. In other words, they diffuse

all over the space in such a way that the probability of finding any agent in any position of space for a sufficiently long simulation time converges to a uniform distribution. It is worth noting that, even though this is true theoretically speaking, in the limit of $t \rightarrow \infty$, we are far away from that limit given that we explore at most 1000 days of simulation time.

Agents were uniformly distributed in the 2-torus space at the start of the simulation. To define proximity between agents, we have divided the 2-torus into $103 \times 103$ square patches of length size equal to 1. This leads to a grid-like space in which both epidemiological interactions and information exchange occur only between agents within the same patch. These interactions happen at rates which are modulated by the spatial density of agents. Thus, the number of total patches was selected to define a rate of interactions so to adjust our ABM to the results of the SEIRD model based on ODEs proposed to explain the EVD dynamics by Weitz and Dushoff [36]. For a deeper dive into the SEIRD model, see S1 Eq. The comparison between the results of our ABM model against the ODE model is shown in S1 Fig.

The epidemiological dynamics of the simulation has two parts: an infection dynamics and a transition dynamics. In general, agents can be in any of the five epidemiological states, i.e. $Q_i \in \{S, E, I, R, D\}$, however, depending on their particular state their dynamics will be of one type or the other. It is worth noting that, $S$ and S represent the susceptible epidemiological state and the susceptible compartment, respectively. The same occurs for the other states and compartments. Importantly, the infection dynamic occurs when a pair of agents susceptible-infected or susceptible-dead is simultaneously found at the same patch. Then, following a probabilistic process, guided by a Montecarlo algorithm, the susceptible agent may suffer a transition from $S$ to $E$, which we will denote from now and on by $S \rightarrow E$, or it may remain in the $S$ state. In practice, when one susceptible agent or multiples susceptible agents encounter either an infected or a death agent in the same patch at the same time, we resolve the Montecarlo step by sampling a random number $v$ from a uniform distribution $\mathcal{U}(0, 1)$. Then, we compare $v$ with $\beta_{I,D} (1 - r_i) < 1$ for each susceptible agent in the patch to resolve the infection: if $v < \beta_{I,D} (1 - r_i)$, then the susceptible agent gets exposed. $\beta_I$ and $\beta_D$ are the infection rates for transitions $S \rightarrow E$ due to the action of $I$ and $D$, respectively, when there is absence of information in the system. These parameters were extracted from reference [36] and they are specific for the 2014–2016 EVD outbreak (these are provided in S1 Table). Of note, the term $\beta_{I,D} (1 - r_i)$ contains the dependency of the *information state* $r_i$. This is actually the modification to the infection rates suggested by Funk el. al [32], and the one we adopt in this article, to couple the communication dynamics to the infection dynamics. Furthermore, since the mean density of agents per patch is approximately 0.94, then the interactions along the simulation are basically pair-wise, this will be also true for the communication dynamics between agents.

On the other hand, the transition dynamic occurs when agents are in any of the epidemiological states but susceptible. In those cases the transitions are $E \rightarrow I$, $I \rightarrow R$, $I \rightarrow D$, and $D \rightarrow \emptyset$. The former and the latter occur in a totally deterministic fashion, after $T_E$ and $T_D$ days respectively. However, $I \rightarrow R$ and $I \rightarrow D$ occur both after $T_I$ days but at rate $1 - f$ and $f$, respectively. The empty state is represented by $\emptyset$, and $D \rightarrow \emptyset$ symbolizes that the agents are being removed from the simulation, which in most of the cases, accounts for a person being buried. All these parameters were also obtained from [36] and are provided in S1 Table.

The *information state* $r_i \in [0, 1]$ accounts for the information that an agent $i$ has about the epidemic at a given time. Whilst $r_i = 0$ indicates that agent $i$ is completely unaware of the epidemic, when $r_i = 1$, agent $i$ becomes completely aware of the epidemic and the state of its environment, knowing exactly how to prevent the infection. We consider $r_i$ as a function of time given by $r_i(t) = \rho_i^{q_i(t)}$, where $\rho_i \in [0, 1]$ is the *awareness decay constant* of agent $i$. On the other hand, $q_i(t) \in \mathbb{N}$ is the *information quality constant* of agent $i$, which is also a function of time,

being $q(t) = 0$ the maximum information quality. Importantly, each agent has its own awareness decay constant that is sampled from a certain distribution $p(\rho)$. This sampling procedure introduces heterogeneity into the system in the sense that every agent may follow a different trajectory along the simulation not only due to the stochastic nature of the Montecarlo simulation, but also influenced by their own awareness. Let us say, we want to apply this model to describe the evolution of an epidemic in a certain society, we state that $p(\rho)$ should be somehow assembled by considering sociological, economic, political, and cultural elements, among others, of that specific society. Nevertheless, to the best of our knowledge, there is no clear theory that could help us to infer the proper $p(\rho)$ based on the actual relationship among all those factors. Still, for simplicity, we can assume that $p(\rho)$ exists, expecting that societies with higher trust, either between each other, the institutions, or the communications media, will have a tendency to have a higher degree of awareness and for longer times, than that of societies with lower trust, particularly when they are exposed to valuable information on how to deal with the spreading of an infectious disease.

The information quality constant $q_i(t)$ has a very interesting dynamics. Counter-intuitively, as mentioned before, maximum information quality is reached when $q_i(t) = 0$, which makes $\beta_{I, D}(1 - r_i)$ to vanish resulting in a form of Behavioral Immune System for that agent, at least during that time step. As stated before, new information can be gathered by agents through three different sources: i) through the action of a global source that periodically feeds information into the system, setting $q_i(t) = 0$ for all the informed agents; ii) from direct contact with agents in the same patch having information of higher quality, setting $q_i(t) = q_j(t) + 1$; or by iii) acquiring the disease, in which case the exposed agent gets informed when receiving the virus, setting $q_i(t) = 0$. Furthermore, information quality degrades in time, one unit per time iteration. Thus, for the case of $t + 1$, i.e. when a time unit goes by in the simulation, and according to the previous paragraph, the information quality constant of an agent $i$ can turn into:

$$q_i(t+1) = \begin{cases} 1 & \text{if either agent } i \text{ is exposed to global information} \\ & \text{or agent } i \text{ is infected at time } t \\ q_j(t) + 2 & \text{if agents } i \text{ and } j \text{ interact and } q_j(t) < q_i(t) \\ q_i(t) + 1 & \text{otherwise} \end{cases} . \quad (1)$$

Of note, when agents $i$ and $j$ interact, information transfer from an agent with higher quality to an agent with lower quality occurs, and simultaneously, a time unit goes by in the simulation. Hence, we add one unit because of communication and one unit because time goes by, resulting in $q_i(t + 1) = q_j(t) + 2$. As expected, communication between agents only occurs if both agents are alive. A schematic representation of our ABM is shown in Fig 1, where we show graphically most of the aforementioned concepts.

To account for the heterogeneity of space in the simulation, agents were located randomly in the grid following a uniform distribution. The initial information quality constant was defined as $q_i(0) = 100$ for all agents, except for the ones that were initially infected, whose initial information quality constant was defined as $q_i(0) = 0$. Despite unbounded, we did set $q_i(0) = 100$ because it is a number representing low information quality such, when applied to $r_i = \rho^{q_i}$, it gives a result close to zero. In other words, when $q_i(0) = 100$, for $\rho \in [0.1, 0.9]$ the information state $r_i \in [10^{-100}, 2 \times 10^{-5}]$, meaning that agents have a negligible awareness of the pandemic situation. To ensure both information and epidemiological dynamics, we choose to start the simulation with 100 agents having $Q_i = I$ and the remaining agents starting in the susceptible compartment, i.e. with $Q_i = S$.

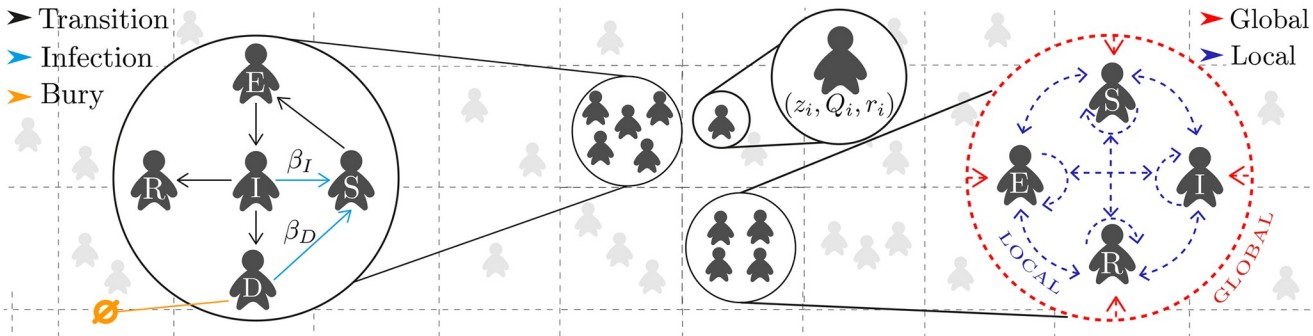

**Fig 1. Graphical scheme of our agent-based model.** The dynamics of the spreading of the infectious disease is depicted in the left side, where $\beta_I$ and $\beta_D$ represent the infection rate of infected agents and dead agents, respectively. Black lines represent deterministic transitions, in days, between states. Blue lines represent infections and, in orange, agents that get buried or removed from the system. The dynamics of information transfer between agents (local communication) is depicted in the right side using blue dashed lines. Information coming from a central entity to the population (global communication), is represented by red dashed lines. Each agent $i$ in the middle zoomed circle is defined by a vector $(z_i, Q_i, r_i)$ where the set of parameters represent its position in the space, its epidemiological state and its information state, respectively. The physical space representation, where each square represent a *patch* and the shaded agents represent a sample of agents distributed in space, is represented by gray discontinued lines in the background.

## Results

### On the influence of homogeneity and heterogeneity of agents

One of the main critiques to traditional ODE-based compartmental models is the restriction imposed by the well-mixed assumption where mass-action dynamics occur. Its equivalence in ABM is when we assume that all agents have the same characteristics, being uniformly distributed in the space, so they are chosen randomly at the moment of updating their epidemiological states. Several authors have discussed this issue suggesting that including heterogeneity is desirable when modeling the dynamics of epidemic outbreaks [32, 69–71]. Heterogeneous and quenched mean-field theories using representations of the space-based on complex networks [72], can also be used to represent heterogeneity in dynamical systems. However, ABM models represent a straightforward way to deal with heterogeneity, since specific features that are unique to each agent can be individually associated. Furthermore, different types of spatial environments with heterogeneous features can be added into the system, for example transit of pedestrians or public gatherings in specific geometries, such as cities or neighbors. Another approach that has shown to be quite successful in this line is the use of complex networks, where nodes can represent either an individual or a group of them. In this context, it has being shown, for instance, that large fluctuations in connectivity between population networks, may strengthen the incidence of epidemic outbreaks [73].

We have in our simulation certain degree of heterogeneity given by the spatial dynamics: agents which start randomly distributed in the space and become spatially closer as the simulation goes by, are more likely to interact than those being distant apart. More importantly, we explored the effect of having homogeneous and heterogeneous societies, by considering both the information quality $q_i$ that each agent has in the system, and different distributions of the awareness decay constant $p(\rho)$. Of note, as $q_i$ changes along simulation time following the interaction dynamics between agents, heterogeneity of information quality is expressed by the different distributions of $q_i$ that may arise from the simulation. On the other hand, as the awareness decay constant $\rho$ is defined as a parameter of the simulation, to enforce heterogeneity we sampled a distribution of $\rho$ considering that $\rho_i \in [0, 1]$. Hence, two options arise: i) the first one representing a homogeneous society, where all agents have the same awareness decay constant $\rho_i = \rho_m$, where $\rho_m$ represents the mode of the distribution. Therefore, the decay

constant for such a society could be formally described as sampling $\rho_i$ from a delta distribution, given by

$$p_1(\rho) = \delta(\rho - \rho_m),\tag{2}$$

where $\delta$ is the Dirac delta and $\rho_m$ represents its center. The second case corresponds to a ii) heterogeneous society, for which the decay constant can be obtained by sampling $\rho_i$ from a truncated Gaussian distribution, given by

$$p_2(\rho) = \frac{\sqrt{2}}{\sigma\sqrt{\pi}} \frac{e^{-\frac{1}{2}\left(\frac{\rho - \rho_m}{\sigma}\right)^2} H(\rho)H(1-\rho)}{\text{erf}[\rho_m/\sqrt{2}\sigma] - \text{erf}[(\rho_m - 1)/\sqrt{2}\sigma]}.\tag{3}$$

In this case, $\sigma$ is the standard deviation of the associated untruncated Gaussian distribution (which was chosen as $\sigma = 0.2$), $\rho_m$ is the mode of the distribution, $H$ is the Heaviside step function, and erf is the Gauss error function. Examples of truncated Gaussian distributions $p_1$ and $p_2$ using different values of $\rho_m$, are shown in panels (a) and (b) from Fig 2, respectively. Importantly, we decided to use the mode of the truncated Gaussian distribution so to produce a distribution of $\rho$ surrounding a central representative value $\rho_m$, which can be understood as a

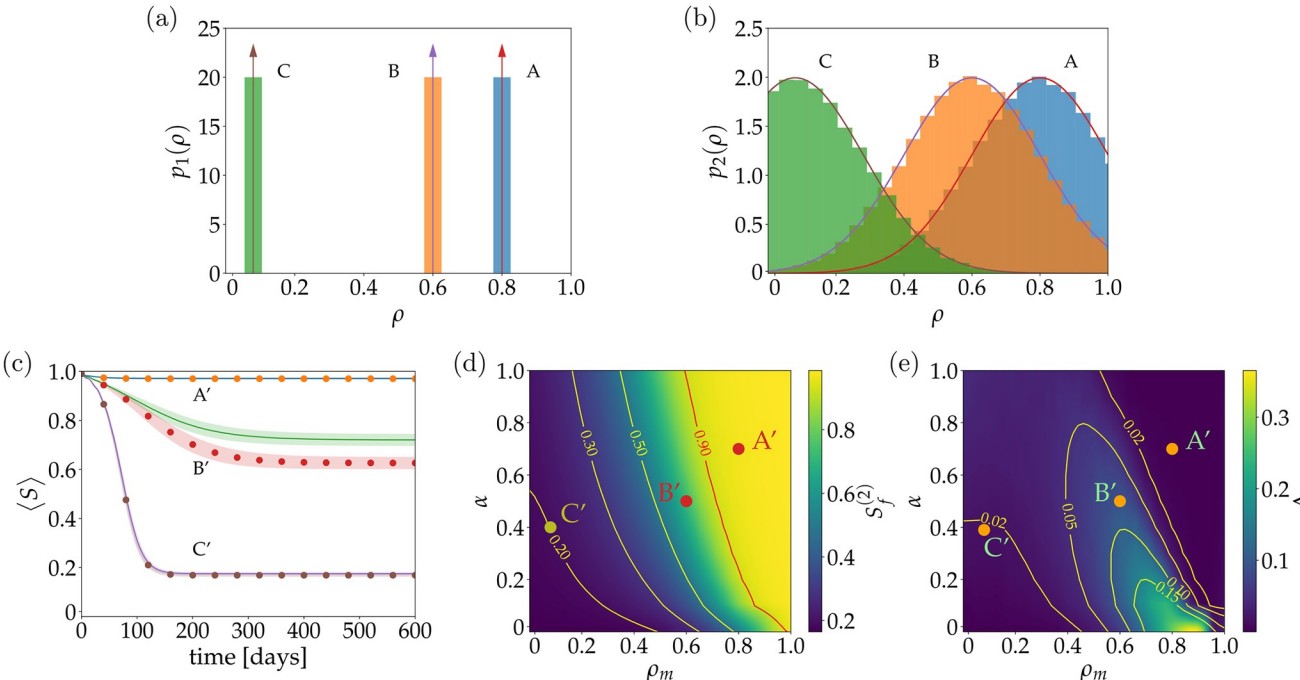

**Fig 2. Accessing homogeneity and heterogeneity effects on the ABM simulation.** Different probability distributions of the awareness decay constant $\rho$ were used to evaluate the effect of heterogeneity by considering a sampling process from a (a) Dirac delta and (b) a truncated Gaussian distribution. In both plots, bars are the sampled distribution and solid lines are the analytic curves, where the arrows in (a) indicate the delta function. (c) Evolution along time of susceptible agents for different values of the mode of decay constant $\rho_m$ vs the ratio of informed agents $\alpha$. Continuous lines represent simulations where $\rho_m$ is sampled from a Gaussian distribution and dotted lines, the sampling from a Dirac delta distribution. Shaded areas represent the standard deviation obtained from 100 independent simulations. (d) Density plot of the final ratio of susceptible agents at the end of an epidemic when there is a heterogeneous distribution of $\rho$, for different values of $\rho_m$ and $\alpha$. Contours lines show the boundary for high and low impact epidemic. (e) Density plot of the difference of final susceptible agents between homogeneous and heterogeneous systems for different $\rho_m$ and $\alpha$. Contour lines delimit areas of low and high difference between systems. In panel (a) and (b), points A, B and C represent different values of $\rho_m$, 0.8, 0.6 and 0.1, respectively. In figures (c), (d) and (e) those points represent the pair $(\rho_m, \alpha)$, being A′ = (0.8, 0.7), B′ = (0.6, 0.5) and C′ = (0.1, 0.4). Points are set in interest regions, high (A′), middle (B′) and low (C′) impact of the information on the epidemic dynamics.

delta distribution with lateral non-symmetrical diffusion, in both directions. As noted in Fig 2b, when producing a truncated Gaussian distribution of $\rho$ using as center the $\rho_m$ obtained from the Dirac delta (Fig 2a), three heterogeneous distributions of $\rho$, denoted A, B and C, were produced. Whereas distribution A resembles a skewed right distribution, distribution C resembles a skewed left distribution.

In order to characterize both the similarities and differences between the homogeneous and the heterogeneous cases, and considering the stochastic nature of the ABM, we executed a large number of simulations, evaluating the effect of different parameters of the system. We used a scheme where information is delivered from a central entity (global information) to a certain portion $\alpha$, randomly selected, of the total population in every step of the simulation. Thus, we have a 2-D parameter space with $(\rho_m, \alpha) \in [0, 1] \times [0, 1]$ and we systematically explore it by sampling $(\rho_m, \alpha)$ with steps $d\rho_m = 0.02$ and $d\alpha = 0.1$. For each point in the parameter space, we run 100 simulations lasting for 1000 simulation days each one, which adds up to a total of 56, 100 independent simulations for each case. We observe that after this extended simulation time, the system has virtually reached its equilibrium state, as can be noted by tracking the evolution of the number of susceptible agents along time. Specifically, we compute $\langle S(t) \rangle$, which is the ratio between the amount of susceptible agents and $10^4$, which is the total number of agents at the beginning of the simulation, at time $t$ and averaged over 100 simulations. A comparison of $\langle S(t) \rangle$ as a function of time for three homogeneous and heterogeneous cases, marked with dotted and continuous lines, respectively, is presented in Fig 2c. Selected points in the parameter space are labeled as A' = (0.8, 0.7), B' = (0.6, 0.5), and C' = (0.1, 0.4). Whilst in A', the disease is stopped right away from a very early stage of the propagation, on the other hand in both B' and C', the disease spreads over the population. We quantify the difference between the outcome of the homogeneous and the heterogeneous cases by computing,

$$\Delta = \left| S_f^{(1)} - S_f^{(2)} \right|, \tag{4}$$

where the superscript (1) and (2) denotes the difference between the homogeneous and heterogeneous cases. For simplicity, from now on we will adopt the notation $S_f^{(i)}$, or simply $S_f$, with suffix $f$ and without the brackets $\langle \cdot \rangle$, to indicates the value of $\langle S(t) \rangle$ at the end of the simulation, i.e. at $t = 1000$ [days]. The use of notation with superscript will be only in sections where multiple cases are discussed at once. In general, $S_f$ is an interesting metric because it allows us to analyze the system at the equilibrium state. Furthermore, $\Delta$ is bounded by 0 and 1 and it represents a difference in terms of fraction of agents relative to $10^4$ (initial total population). Its limiting values indicate that both cases are identical, when $\Delta = 0$, or both cases are totally disparate, when $\Delta = 1$. Remarkably, the outcomes of the simulations in cases A' and C' are quite similar, we have $\Delta < 0.02$, whilst in case B' the difference increases and we have $\Delta \approx 0.07$. Now, we thoroughly explore $\Delta$ over the entire parameter space. In Fig 2d we show how $S_f^{(2)}$ looks as a function of the parameters $\alpha$ and $\rho_m$ in the heterogeneous case. From here we can analyze the effect of the parameters over the outcome of the system. The dark blue zone indicates the region where the disease spread is barely affected by the available information, contrary to what happens in the bright yellow region where the disease is under control. Extreme examples of societies for these two opposite scenarios are those with $\rho_m = 0$ and $\rho_m = 1$, respectively. Additionally, in stark contrast to what our initial intuition could have told us, homogeneous and heterogeneous simulations are very similar for most of the parameter space, as shown in Fig 2e. Here, we can see that the major difference between the simulations is reached when there is very little information being delivered by the central entity, i.e. $\alpha \to 0$, but only in societies with a tendency towards $\rho_m$ closer to 1, but not exactly 1. As $\alpha$ increases, $\Delta$ tends to

decrease, however, this does not happen in a monotonic way for all values of $\rho_m$. Instead, for some values of $\rho_m$ we observe first a subtle increase for small values of $\alpha$ and then a descending behavior.

Overall, these results suggests that the heterogeneous component added by the distribution $p_2(\rho)$ into the simulation is not that relevant, at least in a large domain of the parameter space where its effect is marginal compared with choosing $p_1(\rho)$. It is worth noting that if we could certainly test many distributions to investigate the equivalence of the system, a truncated Gaussian distribution is enough to prove that the system behaves similarly with a slight degree of heterogeneity. With a slight degree of heterogeneity, we refer to a distribution where not heavy tails are present. For this reason, in what follows on this article, we work only in homogeneous systems with $p(\rho) = p_1(\rho)$ as described by Eq (2).

## Non-pharmaceutical strategies based on communications

We now examine the impact of different communication strategies on the epidemic spreading. To do so, we rely on different patterns of information delivery from a central entity to a certain fraction $\alpha$ of the population. Hence, we study three different strategies, namely strategy I, II, and III. By following strategy I, we inform on a daily basis to a fraction $\alpha$ of the agents in the simulation, that is chosen randomly among the living agents, following a uniform distribution. In strategy II, we inform to the entire population, but contrary to the previous strategy, this happens every $\tau$ days. Finally, in strategy III, we inform on a daily basis to the entire population, only after $\delta$ days have passed, since the beginning of the epidemic.

As we previously proceeded, in all three strategies we are interested in the outcome of the epidemic by evaluating the number of susceptible agents at the end of the simulation, *i.e.* at $t = 1000$ [days]. However, at $t = 600$ [days] the system has virtually reached the equilibrium state. In this section, we use a slightly different notation compared to that of the one used in the previous section. The superscript $(x)$ in $S_f^{(x)}$ can take values in $\{\alpha, \tau, \delta\}$ and it now indicates which strategy is being analyzed. Again, depending on the context we use either $S_f^{(x)}$ or simply $S_f$.

**Strategy I: Daily information to a fraction $\alpha$ of the population.** The first strategy, discussed in general terms in the previous section, will be analyzed in further details. As seen in Fig 3a, the value of $S_f^{(\alpha)}$ is depicted for 11 simulations executed considering different values of $\alpha$, ranging from 0 to 1. We also present a control case, indicated by a dashed line at the bottom and a double asterisk symbol (\*\*), for which no information at all is present in the system. In this case, we obtain a value of $S_f \approx 0.16$. We may recall that there are three mechanisms for the agents to acquire information: i) from agent to agent; ii) when agents acquire the disease and get informed, and the last one, iii) by agents acquiring information from a central entity. In the case of Strategy I, the first two mechanisms are always on, therefore, even when $\alpha = 0$, agents may still have access to new information. Of note, the latter case may act as a control situation in which $\alpha = 1$. We termed this case as the *ideal* situation considering the complexity involved in implementing such a strategy during a real epidemic—accounting for logistic and other factors. As expected, the ideal case has the best results in terms of reducing the impact of the epidemic for all values of $\rho_m$. This case is highlighted with a single asterisk symbol (\*) and appears on top of all the other more realistic cases. It is worth noting that, both controls, i.e. cases with single asterisk and double asterisk, appear inherently for the three strategies, as shown in panels (b) and (c) of Fig 3.

As expected, all curves $S_f$ in Fig 3 suffer a transition from a state where communication does not affect the epidemic spreading at all, for values of $\rho_m$ closer to 0, to a state where

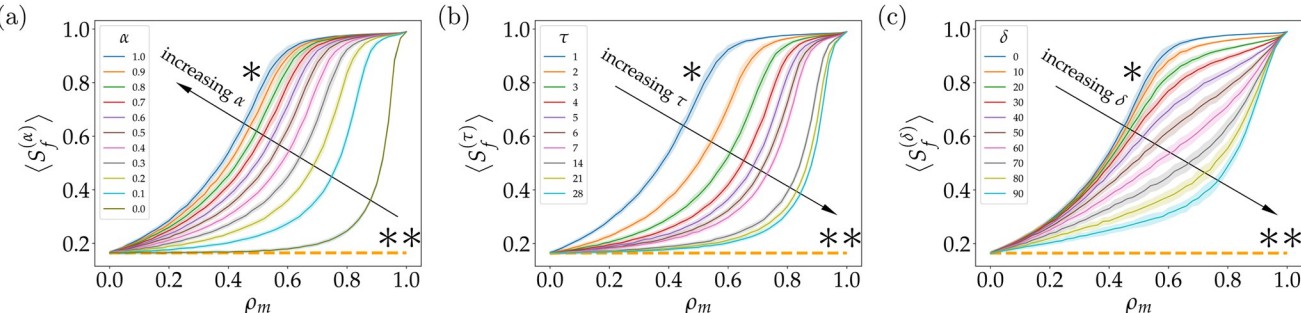

**Fig 3. Accessing the effect of different communication strategies on the epidemic outcome.** In all panels the final ratio of susceptible agents after 1000 days of simulation for different values of $\rho_m$. (a) Central information delivered to the population while the ratio of informed agents changes. Different curves represent different ratios of informed agents $\alpha$, ranging from 0 (olive line) to 1 (blue line) with 0.1 interval. (b) Periodicity of information delivery. Different curves represent different periodicity of information $\tau$, delivered to the population. We have tested periodicity daily from 1 day (blue line) to 7 days (pink line), and then weekly, at 14, 21 and 28 days (green line). (c) Delay in starting information delivery. Different curves represent different delays $\delta$, considering the elapsed time to deliver the message from the beginning of the epidemic. We have tested delay in the first message from 0 (blue line) to 90 days (light blue line) with 10 days interval. After the first message arrives, subsequent information is delivered daily. In all panels, shaded areas represent the standard deviation for 100 simulations. A $*$ above blue line indicates the ideal strategy, and $**$ above orange dashed line indicates the worst strategy, *i.e.*, when there is no central information delivered to the population neither information delivered to infected agents. Curves with higher values of $\langle S_f^{(x)} \rangle$ where $x \in \{\alpha, \tau, \delta\}$ implied a better strategy result, *i.e.*, less infected agents.

communication conduces to a complete suppression of the epidemic, stopping its spreading, for values of $\rho_m$ closer to 1. The way how these transitions occur for different values of $\alpha$, however, is quite interesting and highly nontrivial. In all cases, we observe a type of generalized logistic behavior in the transitions. As a consequence of this transition, the effectiveness of the strategy depends strongly on the characteristic $\rho_m$ of the society and, of course, the parameter $\alpha$. For example, for societies with $\rho_m = 0.8$ we have that if no actions are taken in terms of informing agents from any central entity, that means $\alpha = 0$, then the effect is quite similar compare to the control double asterisk (see Fig 3). On the contrary, if we inform as little as $\alpha = 0.1$ of the total population, the increment in the amount of susceptible agents at the end of the simulation is considerable, having $S_f \approx 0.59$. When $\alpha = 0.3$ we basically stop the epidemic from the beginning and $S_f \approx 0.92$ in this case, which is remarkable considering the small amount of agents being informed. Nonetheless, if we perform the same analysis but this time we consider $\rho_m = 0.2$, the impact of the strategy is very limited, even if we have $\alpha = 1$, which is the ideal case. The scenario at the end of the epidemic, in this case, is $S_f \approx 0.28$. In a more intermediate case, such as considering a society with $\rho_m = 0.5$, the results are more sensitive to the variation of $\alpha$ over a larger range of parameters. In this case, for example, we can jump from $S_f \approx 0.34$, for $\alpha = 0.3$, to $S_f \approx 0.51$, for $\alpha = 0.6$, or to $S_f \approx 0.69$, for $\alpha = 0.9$. As noted in Fig 3a, there is an evident non-linear relationship between $\alpha$ and $\rho_m$. Particularly, we can see that our system behaves similarly for different values of $\alpha$ when $\rho_m$ is close to 0.0 or close to 1.0, but for intermediate values of $\rho_m$, the high non-linearity of the system arise, denoted by a higher difference for the $\alpha$ curves. When $\rho_m$ is close to 0.0, all the population has a fast decay of awareness, and inversely, when $\rho_m$ is close to 1.0 all the population have a slow decay of awareness, so the percentage of informed people $\alpha$ does not contribute substantially to the final size of the epidemic. Now, for an intermediate case of $\rho_m$, there is not a clear predominance of the agents with fast decay of awareness or agents with slow decay of awareness, resulting in a substantial difference when informed different fraction of population $\alpha$.

This high variability in the results depending on the election of $\rho_m$ is not uncommon at all in these type of studies. The inference of the right parameters is one of the most difficult parts

when trying to apply these type of models in a real situation. For this reason, the use of empirical data to feed models should be mandatory when a more concrete or applied analysis is required. However, everything is not lost and we can still use these models to learn valuable lessons and generate intuition.

**Strategy II: Information every $\tau$ days to the entire population.** We now turn to assess the impact that the periodicity of sending information could have on the spread of an infectious disease. In this strategy we have started informing on day 0, regarding the epidemic start day, and varied the number of elapsed days between consecutive messages. We started with a low periodicity, daily information delivery, and then we changed one day a time until completing the first week, then we tried with 2, 3 and 4 weeks of delay between messages from a central entity.

Results from this strategy are shown in Fig 3b, where we can see how the distances between curves $S_f$ decrease as we increase the period in which messages to the population are sent. We also highlight the difference between informing daily ($S_f$ shown by *) and informing each two days (orange curve), where we can notice the largest difference between consecutive $S_f$. We can see this, for example, in societies with $\rho_m = 0.5$, where if we have daily information delivery $\tau = 1$ then $S_f \approx 0.75$, but if we increase the delay in one day, *i.e.*, $\tau = 2$, then $S_f \approx 0.50$, which indicates a difference in the final susceptible population of 0.25 between informing daily and each two days. If we see $\tau = 3$, we have that $S_f \approx 0.38$, which is a difference in the final susceptible population of 0.12 comparing to $\tau = 2$, which is less than half of the difference between $\tau = 1$ and $\tau = 2$. This result shows us that, as the periodicity of information increases, it becomes less relevant to delay or to advance one day the information delivery, being the largest difference between informing daily and each two days. To elaborate from another point of view on this conclusion, we include S2 Fig, where the final number of susceptible agents changes when the timescale of information, represented by $\tau$, varies. Indeed, we can see in S2 Fig a fast decay for the firsts $\tau$, and then a slow decay, representing a kind of "long-tailed" decay, which denotes the importance of sending information with low periodicity—*i.e.* high frequency.

In literature, we find a similar work [74], that also focuses on the effects of sending periodic information to the population, denoted by authors as "pulsating campaigns", under the assumption that there is a communication of the *risk* towards people, which fulfills a role similar to awareness in our work. Their conclusion regarding this type of campaign is that it is better a pulsating campaign instead of a campaign where the people are informed constantly, all this under an oscillatory dynamics of infection, *i.e.* where the infected cases grow, decay, and then repeat this dynamic. They explain this result as a consequence of an abrupt increase in risk communication when starting a campaign. Despite is not possible a direct comparison, as we don't have oscillatory dynamics of infection, we can make some assumptions, for instance, that if we had oscillatory infection dynamics, we probably wouldn't have some abruptly increase of awareness (doing a simile with risk definition), and due to this, contrary to [74], it is probable that a continuous strategy would be better in our model (shorter periods are better for the population in our model). Without a doubt, delving into these types of strategies can show us optimal ways to reduce infections.

In summary, our results show that the best strategy to achieve a significant reduction in the number of infected agents at the end of the simulation, is to inform daily. Although this result seems obvious, we have shown that the impact of informing with a period larger than one day is very severe, conducing to a dramatic increment in the amount of infected agents at the end of the simulation for most of the scenarios.

**Strategy III: Daily information to the entire population after $\delta$ days.** Another interesting question we want to address is: how much the delay in implementing the information strategy may affect the impact of the epidemic? In other words, we would like to figure out what happens when the information strategy includes a delay between the starting of the

epidemics and the delivery of information to the population. To answer this question we changed from 0 to 90 days, each 10 days, the start day of the information delivery. To be consistent with the previous findings, after the information strategy started, we delivered information on a daily basis. Results from this strategy are shown in Fig 3c, where considering a delay of $\delta = 0$ and $\delta = 10$ the resulting curves are very close, indicating that it is similar to start the information strategy on day zero, compared to starting it 10 days after the start of the epidemic. This effect becomes even clearer when comparing three different societies, *i.e.* three different $\rho_m$, through its different curves $S_f$. To do so, we evaluated the curves resulting from a delay of $\delta = 0$ and $\delta = 10$. For $\rho_m = 0.2$ there is a difference $<0.01$ in the ratio of final susceptible agents, when $\rho_m = 0.5$ the curves have a difference of 0.03, and when $\rho_m = 0.8$ the difference is 0.02. As noted, all the numeric values are close to zero, which tells us how similar is the outcome of all these strategies. A similar situation becomes evident if we compare the curves for $\delta = 0$ and $\delta = 20$, where for the case of $\rho_m = 0.2$ the difference is $<0.01$, while for the case of $\rho_m = 0.5$, the difference is 0.07, and for the case of $\rho_m = 0.8$, the difference is 0.04. These results show us that for $\delta = 20$ there is a subtle difference comparing to the ideal case of $\delta = 0$. Now, instead of comparing curves of delays $\delta$, we compare different $\delta$ for the same society. For societies with low values of $\rho_m$, for example $\rho_m = 0.2$, the final population of susceptible agents $S_f$ is similar when applying different delays $\delta$ in delivering the information: when $\delta = 30$, a final susceptible population of $S_f \approx 0.25$ is obtained, and when $\delta = 60$, a $S_f \approx 0.28$ is obtained, whereas for the case of $\delta = 90$, a value of $S_f \approx 0.29$ is obtained. These results highlight how similar are the responses of the system when considering different delays in information delivery. Of note, this difference become larger when we have societies with higher $\rho_m$, for example $\rho_m = 0.5$, where we have that for the cases of $\delta = 30$, $\delta = 60$ and $\delta = 90$, the final susceptible population is $S_f \approx 0.45$, $S_f \approx 0.64$ and $S_f \approx 0.75$, respectively.

With these results in mind, we may conclude that the delay in information delivery has less impact in the size of the epidemic for societies having $\rho_m$ in the extremes, *i.e.* low and high values of $\rho_m$.

When comparing these results with that of strategy II in which we evaluate the effect of the periodicity of information, we may argument that is more important to inform on a daily basis than starting the promptly, as-soon-as-possible, delivery of information to the population. It is worth noting that this result is consistent with the one presented in [22], where the authors discuss about the importance of the rate at which awareness programs are executed. This same strategy comparison is included in S3 and S4 Figs but from the point of view of cumulative dead agents and removed agents, respectively, which, in addition to reinforcing this conclusion, could help to better understand the impact of these results. For completeness, in the next subsection we proceed to compare the outcome of these three strategies.

**Accessing the similarities between strategies.** An interesting observation from Fig 3 is that several curves $S_f$, as a function of $\rho_m$, share a similar shape between them for different strategies. This leads to an interest from our part in quantifying this similarity and try to establish a kind of equivalence relation, between specific pairs obtained from different strategies. For example, by simple visual inspection one could be tempted to say that informing every $X$ days to the entire population is more or less equivalent than informing every day to a $Y$ portion of the total population. However, this equivalence immediately raises various questions. For instance, a fundamental one is how can we say that two curves are equivalent, or similar, in this context? To address this interesting question, we used a measure of the distance between two curves $S_f^{(x)}$ and $S_f^{(y)}$ given by

$$\zeta(x,y) = \int_0^1 \left| S_f^{(x)} - S_f^{(y)} \right| d\rho_m , \tag{5}$$

where $x$ and $y$ symbolize the value of the parameters for the different strategies that we are comparing. To illustrate this, let us say that we are interested in comparing strategy II with strategy I for the parameters $\tau = 2$ and $\alpha = 0.6$. In this case, we have $\zeta(2, 0.6) = 0.17$ where this number indicates the area between the two curves. The closer $\zeta$ is to zero, the closer the curves are to each other. We have observed that when $\zeta$ is less than 0.35 the two curves are virtually indistinguishable from each other and each of them is within the standard deviation of the other for almost all of the values of $\rho_m$. We are now interested in solving a minimization problem for $\zeta$, in which given a certain value of the parameter $x$ for one strategy, we are interested in finding the value of the parameter $y$ for another strategy, such that it minimizes $\zeta$, i.e. there is no other value of $y$ that gives a lower value for $\zeta$. Some examples of pair of curves that minimize $\zeta$ are shown in S5 Fig for different strategies. Furthermore, we explore this in a systematic way and find a large collection of pairs $(\tau, \alpha)$, $(\tau, \delta)$, and $(\alpha, \delta)$, which are summarized in Table 1. Interestingly, when comparing strategy II with strategy I, we can find very similar results for the outcome of the epidemic for a large collection of parameters—for all the values of $\tau$ that we explored we were able to find a value of $\alpha$ such that it minimizes $\zeta$ and also holds the condition $\zeta(\beta, \alpha) < 0.35$. Therefore, we conclude that, independently on the value of $\rho_m$ for the society and under the assumptions of our simulations, informing to the entire population every 2 [days] conduces to similar results than informing to just a 0.60 of the population every day. On the same page, informing to the entire population every 3 [days] conduces to similar results than informing to just a 0.38 of the population every day. And so on, following the values in Table 1. We would like to highlight now a point that is extremely valuable, and definitively it is something that should be worth to take into consideration when exploring communication strategies in a more practical scenario: $\alpha$ decays rather quickly as $\tau$ increases, as shown in Table 1. This remarks the notorious importance of keeping a well-informed population under a quite frequent scheme during an epidemic.

When comparing $\tau$ with $\delta$, even though we can still formally solve the minimization problem, we find that the curves are quite different, at least for a broad range of the parameter $\rho_m$. This is shown in Table 1 by the large values obtained for $\zeta$, which are all above 1.3, except for the control cases $\tau = 1$ and $\delta = 0$. Something very similar to this happens when comparing $\delta$ with $\alpha$, however, additionally to the control case, we also have a single nontrivial case with $\delta = 10$ and $\alpha = 0.96$ that provides $\zeta(10, 0.96) = 0.33$.

**Table 1. Similarity between different communication strategies.**

| $\tau$ | $\alpha$ | $\zeta(\tau, \alpha)$ | $\tau$ | $\delta$ | $\zeta(\tau, \delta)$ | $\delta$ | $\alpha$ | $\zeta(\alpha, \delta)$ |
|---|---|---|---|---|---|---|---|---|
| 1 | 1.00 | 0.07 | 1 | 0 | 0.08 | 0 | 1.00 | 0.09 |
| 2 | 0.60 | 0.17 | 2 | 28 | 1.63 | 10 | 0.96 | 0.33 |
| 3 | 0.38 | 0.12 | 3 | 60 | 2.07 | 20 | 0.90 | 0.67 |
| 4 | 0.26 | 0.14 | 4 | 74 | 1.96 | 30 | 0.82 | 1.03 |
| 5 | 0.20 | 0.08 | 5 | 80 | 1.79 | 40 | 0.68 | 1.44 |
| 6 | 0.16 | 0.15 | 6 | 86 | 1.57 | 50 | 0.52 | 1.78 |
| 7 | 0.12 | 0.12 | 7 | 90 | 1.38 | 60 | 0.42 | 1.93 |

A comparison between the three strategies of information delivery evaluated in this work is presented above, organized in subtables. On each subtable, values from the first and second column are compared. The third column presents the $\zeta$ value corresponding to the area under the curve of the absolute value computed for the difference between curves $S_f^{(x)}$ and $S_f^{(y)}$, corresponding to the compared strategies (see Eq (5)). Close values of $\zeta$ to zero are indicative of more similar curves.

## On the impact of agents's communication in the spread of an infectious disease

Now we turn to explore the importance of communication between agents on the impact of the spread of an epidemic disease. To do so, we evaluated the impact of strategies focusing on encouraging agents's communication to decrease the size of an epidemic. Hence, we consider communication between agents as the act of sharing information about the disease between agents, when they are physically close to each other.

**Assessing the impact of agents's communication without the delivery of central information.** First, we explore the impact on the epidemic when there is only communication between agents, without central information delivery. In other words, agents do not receive information from mass media/central entity, neither infected agents receive information when acquiring the virus. Therefore, the only source of information is the initial information that agents have at the beginning of the simulation. We have tested this strategy to determine if solely the communication between agents is capable to stop the spread of an infectious disease. To do so, we set the system with ratios 1, 0.1, 0.01 and 0.001 of the initial number of informed agents, allowing agents to communicate freely between them. Results in Fig 4a, denote that the impact on the epidemic outcome of this strategy is meaningless, as the four curves appear overlapped. In other words, these results tell us that no matter how many agents are informed at the beginning of the epidemic, when central information is absent, the final size of the epidemic will not be affected.

Although is expected that this strategy is not effective, due to the fast decay of information quality in time, given by $q_i(t + 1) = q_i(t) + 1$, what it is surprising is that it has zero impact even when all population is informed at the beginning of the simulation. This result shows us that the initial information of the system has no impact on the output of an infectious disease when there is no central information being delivered to population. Of note, we may also say that a strategy relying purely on agents's communication should not be enough to stop the spread of the disease, for most of the values of $\rho_m$.

**Encouraging communication between agents when there is central information delivery.** Finally, we evaluated the impact on the dispersion of the epidemic when both agents to agents communication and central information are present. To do so, we compare the results of systems where no communication between agents is present with that of the system when communication between agents is activated. In all systems we included the delivery of information to agents once they acquire the virus have considered information to recently infected agents (they obtain information when they get infected). To illustrate the difference between these systems we have selected three points A, B and C that belong to the parameter space ($\rho_m$, $\alpha$), as can be seen in Fig 4b. For each selected point, two curves are depicted: the continuous curve representing deactivated communication between agents and the dotted curve, representing activated communication between agents. We can see that for A and C the continuous and dotted curves are identical, but for B there are some differences, showing us that, in this point, communication between agents actually affects the output of the spread of the infectious disease. To further study this behavior, exploring in a systematic way the parameter space formed by ($\rho_m$, $\alpha$), we created a density plot of $\rho_m$ and $\alpha$, as seen in Fig 4c. In panel c), the difference between systems can be further accessed by relying on the color bar map. We can see that for a big portion of the space, communication between agents is almost irrelevant (dark blue zones), but there is a portion where it plays an important role, being A and C in the zone where agents communication is irrelevant and B being in the zone showing relevant differences. This interest zone where agents communication plays an important role is characterized for a high $\rho_m$ and a low $\alpha$, which could be interpreted as when there is little to non central

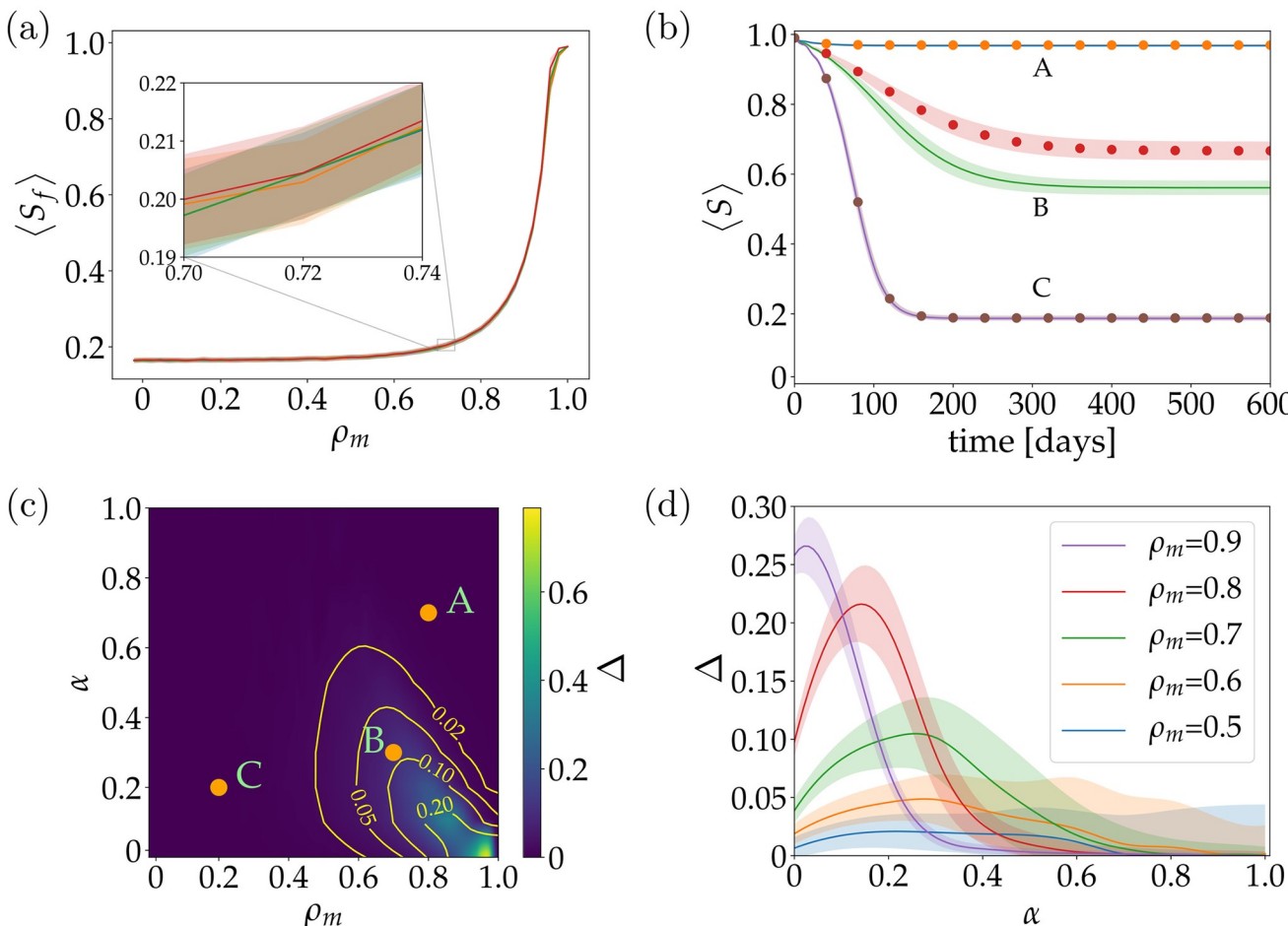

**Fig 4. Assessing the impact of communication between agents on the outcome of an epidemic.** (a) Only agents to agents communication without central information in the system is allowed, considering different proportions of the initial population informed: 1, 0.1, 0.01 and 0.001. The four curves represent the final ratio of susceptible agents of a 600 [days] simulation when the mode of the awareness decay constant $\rho_m$, increases. Shaded areas represent the standard deviation for 100 simulations. The inset figure zooms in the interval $\rho_m \in (0.70, 0.74)$ to graphically show how close to each other the curves actually are. (b) Evolution in time of susceptible agents for different values of $\alpha$ and $\rho_m$ when central information is available. Continuous lines represent simulations where communication between agents is inactivated and dotted lines where communication is activated. (c) Density plot of the difference between systems where agents communication is inactivated and activated for different values of $\rho_m$ and $\alpha$. For figures (b) and (c) the points A, B and C represent the pair ($\rho_m, \alpha$) with values (0.8, 0.7), (0.7, 0.3) and (0.2, 0.2), respectively. (d) Difference between two systems, where in the first one agents communication is inactivated and the second one is activated, for different ratio of informed agents $\alpha$. Different curves represent different $\rho_m$. Shadows areas represent the propagation of uncertainty (both systems have a standard deviation resulting from sampling 100 simulations, and because of this, we propagate the uncertainty of the difference between the systems).

information delivery (low values of $\alpha$) and societies with a tendency towards $\rho_m$ closer to 1, the communication between agents could help to decrease the size of an epidemic. Likewise, when there is medium to high central information delivery, this being around $\alpha = 0.45$ and above, communication between agents becomes irrelevant, independently of the $\rho_m$ of the society. For completeness, we show in more detail the zone of interest, ranging from $\rho_m = 0.5$ to $\rho_m = 0.9$ and $\alpha = 0.0$ to $\alpha = 1.0$, in Fig 4d where we can see how the difference between systems with agents communication activated and deactivated change due to different $\rho_m$ and $\alpha$. We can clearly see a tendency that for higher $\rho_m$ and lower $\alpha$, larger is the difference between systems, decreasing when $\alpha$ increases. This again, show us that communication between agents become irrelevant when there is a high central delivery of information to population. As we are not

capable, yet, to know the $\rho_m$ for a society, we think that this strategy of encouraging agents communication should always be implemented, because if we are lucky enough of being implementing this strategy to a society with $\rho_m$ close to 1, we will have a considerable impact in reducing the size of the epidemic. We think that this strategy of encouraging agents communication could be useful when there is limited access to public or massive information, as was the case of the 2014–2016 Ebola outbreak in West Africa, where the affected countries had limited access to mass communication [75].

## Conclusion

Since the beginning of 2020 the world has been living under the influence of the COVID-19 pandemic. Hence, we dealt with the idea of using COVID-19 instead of EVD to study the influence of information on the spreading of an infectious disease. Despite the abundant COVID-19 related literature, to the best of our knowledge, still no consensus on both models and parameters successfully describing the COVID-19 dynamics has been reached [76–78]. Therefore, we decided to stick with EVD because the SEIRD model, using proper parameters, has proven to effectively describe the evolution of EVD in human populations [36].

From a modeling perspective, EVD as well as other infectious diseases, share similar spreading principles that can be captured using compartmental models. Even though we presented results coupling information with the SEIRD model, this coupling could be possible using any other compartmental model. In this sense, our approach is somehow universal. For instance, if we wanted to apply our model to COVID-19, despite exhibiting a SEIRD-compatible dynamics, then the parameters should be set to the specific values of COVID-19 instead of EVD.

To study the influence of different communication strategies on the spreading of infectious diseases such as EVD, we decided to use the ABM framework. This framework allows us to explore at an individual level, the coupling of information and the spreading of an infectious disease, evaluating the influence of global (i.e. central entity or mass media) and local (i.e. agents' communication) information, assuming that awareness decays in time and information's quality decays when it is transferred from one person to another.

Our results show a remarkable difference in the impact of the epidemic when we compare a population not informed at all against a small fraction of informed population. Of note, our results show that it is preferable to have a communication strategy delivering daily messages than the delivery of prompt information, as soon as possible. Considering the dynamics of local communication, we show that the initial number of informed agents is irrelevant to the output of an epidemic when new information is not entering frequently into the system. Moreover, for some societies, local communication between agents plays an important role when the information entering into the system is scarce, becoming irrelevant when a large portion of information is regularly coming from a central entity.

Regarding how our results can be associated with certain societies, we postulate that when $\rho_m$ becomes close to 1, it should resemble societies with high trust. Of note, high trust societies tend to exhibit higher social capital, a beneficial characteristic that may lead not only to economic growth [79] but also to the effective suppression of the spread of an infectious disease. Probably, the most notable case is the Zero COVID strategy implemented by New Zealand and Australia [80], two countries that exhibit high trust [81]. On the other hand, a $\rho_m$ close to 0, should represent societies with low trust such as Argentina [81]. In this case, the impact on the size of the epidemic depending on the information delay is low due to the lack of agents trust in the central authorities.

As a whole, our results quantify the impact of communication strategies in the spread of an infectious disease and also show the equivalences between, at first sight, different approaches

considering both sources and frequency of delivery. We also paid emphasis to the role that communication between agents may play to determine the final size of an epidemic.

An interesting unaddressed question that we will pursue in future work is to determine the influence of misinformation, or *fake news*, on the outcome of the pandemic. Previous works have shown that misinformation and information have notoriously different dynamics [64]: misinformation spread faster and farther than information. Studying the behavioral modulation induced by misinformation could lead us to determine its effects on the population's awareness and hence, its impact on the dispersion of infectious diseases.

As a whole, this work helps to understand the effectiveness of communication strategies as non-pharmaceutical interventions to tackle epidemics. This knowledge can be lead to the implementation of evidence-based public policies focused on the adoption of preventive behaviors, which could ultimately lead to saving human lives.

## Supporting information

**S1 Eq. SEIRD model equations.** Equations that describe the evolution in time of the states of the system, where $S$ are susceptibles, $E$ exposed, $I$ infected, $R$ removed and $D$ dead agents. The parameters of the system are shown and explained in S1 Table. This model was proposed by [36].
(PDF)

**S1 Fig. ABM evaluation against the literature.** (a) ABM, temporal evolution of the states of the system for 100 repetitions. The shaded area is the standard deviation. (b) Same model in deterministic ODEs system. In both panel, curves represent the temporal evolution of the states of the system. The legend for each states represent the final ratio of agents at the end of the simulation. Maximum infected in ABM, 577 persons in $t = 129$ [days], maximum infected in ODE, 542 persons, in $t = 135$ [days]. It is worth noting that these systems do not include the information model".
(TIF)

**S2 Fig. Normalized area under the curve $\Lambda$ vs $\tau$.** $\Lambda$ represent the area under the curve for each curve $\tau$ in Fig 3b. We can see how $\Lambda$, represented by red dots, decrease as $\tau$ increase. (a) Daily analysis. (b) Weekly analysis. Continuous lines represent the tendency of the curve.
(TIF)

**S3 Fig. Accessing the effect of different communication strategies on the epidemic outcome for cumulative dead agents.** We replicate the analysis in Fig 3 but instead of susceptible agents now we have the cumulative dead agents. All panels show the final ratio of cumulative dead agents after 1000 days of simulation for different values of $\rho_m$. (a) Central information delivered to the population while the ratio of informed agents changes. Different curves represent different ratios of informed agents $\alpha$, ranging from 0 (olive line) to 1 (blue line) with 0.1 interval. (b) Periodicity of information delivery. Different curves represent different periodicity of information $\tau$, delivered to the population. We have tested periodicity daily from 1 day (blue line) to 7 days (pink line), and then weekly, at 14, 21 and 28 days (green line). (c) Delay in starting information delivery. Different curves represent different delays $\delta$, considering the elapsed time to deliver the message from the beginning of the epidemic. We have tested delay in the first message from 0 (blue line) to 90 days (light blue line) with 10 days interval. After the first message arrives, subsequent information is delivered daily. In all panels, shaded areas represent the standard deviation for 100 simulations. A $*$ under blue line indicates the ideal strategy, and $**$ below orange dashed line indicates the worst strategy, *i.e.*, when there is no central information delivered to the population neither information delivered to infected

agents. Curves with lower values of $\langle D_c^{(x)} \rangle$ where $x \in \{\alpha, \tau, \delta\}$ implied a better strategy result, *i.e.*, less dead agents.
(TIF)

**S4 Fig. Accessing the effect of different communication strategies on the epidemic outcome for removed agents.** We replicate the analysis in Fig 3 but instead of susceptible agents now we have removed agents. All panels show the final ratio of removed agents after 1000 days of simulation for different values of $\rho_m$. (a) Central information delivered to the population while the ratio of informed agents changes. Different curves represent different ratios of informed agents $\alpha$, ranging from 0 (olive line) to 1 (blue line) with 0.1 interval. (b) Periodicity of information delivery. Different curves represent different periodicity of information $\tau$, delivered to the population. We have tested periodicity daily from 1 day (blue line) to 7 days (pink line), and then weekly, at 14, 21 and 28 days (green line). (c) Delay in starting information delivery. Different curves represent different delays $\delta$, considering the elapsed time to deliver the message from the beginning of the epidemic. We have tested delay in the first message from 0 (blue line) to 90 days (light blue line) with 10 days interval. After the first message arrives, subsequent information is delivered daily. In all panels, shaded areas represent the standard deviation for 100 simulations. A $^*$ under blue line indicates the ideal strategy, and $^{**}$ below orange dashed line indicates the worst strategy, *i.e.*, when there is no central information delivered to the population neither information delivered to infected agents. Curves with lower values of $\langle R_f^{(x)} \rangle$ where $x \in \{\alpha, \tau, \delta\}$ implied a better strategy result, *i.e.*, less agents that were infected and then removed.
(TIF)

**S5 Fig. Differences between communication strategies.** Here we show some examples of how similar is one strategy compared to another, where $\zeta_{xy}$ denotes the differences between curves $x$ and $y$. (a) Comparison of informing each 3 days the whole population and the 38 percent daily. (b) Comparison of informing each 21 days the whole population and the 2 percent daily. (c) Comparison of informing with a delay of 20 days the whole population and to inform daily the 90 percent of population. (d) Comparison of informing with a delay of 30 days the whole population and to inform daily the 82 percent of population. (e) Comparison of inform each 6 days the whole population and with a delay of 90 days. (f) Comparison of inform each 3 days the whole population and with a delay of 60 days.
(TIF)

**S1 Table. Parameters for the SEIRD model.** This values are for 2014–2016 Ebola outbreak in West Africa, being $T_E$ time that spend a person in Exposed state before become Infected, $T_I$ time that spend a person in Infected state before Dead or Recover, $T_D$ time that spend a person in Dead state before it get buried, $f$ fraction of infected individual that die, $\beta_I$ the infection rate of Infected to Susceptible person and $\beta_D$ the infection rate of Dead to Susceptible person. These parameters were extracted from [36].
(PDF)

## Author Contributions

**Conceptualization:** Alejandro Bernardin, Alejandro J. Martínez, Tomas Perez-Acle.

**Data curation:** Alejandro Bernardin.

**Formal analysis:** Alejandro Bernardin, Alejandro J. Martínez.

**Funding acquisition:** Tomas Perez-Acle.

**Investigation:** Alejandro Bernardin, Alejandro J. Martínez.

**Methodology:** Alejandro Bernardin, Alejandro J. Martínez, Tomas Perez-Acle.

**Project administration:** Tomas Perez-Acle.

**Resources:** Tomas Perez-Acle.

**Software:** Alejandro Bernardin, Alejandro J. Martínez.

**Supervision:** Alejandro J. Martínez, Tomas Perez-Acle.

**Validation:** Alejandro Bernardin.

**Visualization:** Alejandro Bernardin, Alejandro J. Martínez.

**Writing – original draft:** Alejandro Bernardin, Alejandro J. Martínez.

**Writing – review & editing:** Alejandro Bernardin, Alejandro J. Martínez, Tomas Perez-Acle.

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
