## [Decision Letter · Decision Letter 0]

29 Apr 2021

PONE-D-21-09046

On the effectiveness of communication strategies as non-pharmaceutical interventions to tackle epidemics

PLOS ONE

Dear Dr. Martinez,

Thank you for submitting your manuscript to PLOS ONE. After careful consideration, we feel that it has merit but does not fully meet PLOS ONE’s publication criteria as it currently stands. Therefore, we invite you to submit a revised version of the manuscript that addresses the points raised during the review process.

We look forward to receiving your revised manuscript.

Kind regards,

Sebastián Gonçalves, Ph.D.

Academic Editor

PLOS ONE

Journal Requirements:

Reviewers' comments:

Reviewer's Responses to Questions

**Comments to the Author**

1. Is the manuscript technically sound, and do the data support the conclusions?

Reviewer #1: Yes

Reviewer #2: Yes

2. Has the statistical analysis been performed appropriately and rigorously? 

Reviewer #1: Yes

Reviewer #2: Yes

3. Have the authors made all data underlying the findings in their manuscript fully available?

Reviewer #1: Yes

Reviewer #2: No

4. Is the manuscript presented in an intelligible fashion and written in standard English?

Reviewer #1: Yes

Reviewer #2: Yes

5. Review Comments to the Author

Reviewer #1: Comments for PONE-D-21-09046

This paper evaluates the effectiveness of social and communication factors within the population during an epidemic outbreak. The authors use an agent-based susceptible-exposed-infected-recovered-dead (SEIRD), epidemiological model, focusing on simulation results to reach a series of conclusions about the effectiveness of population communication. The model is meaningful and the derivation of the main results sounds right. But it needs to be revised before the second-round consideration. Here are some comments for this manuscript.

The summary says, “Moreover, having a scheme of delivering daily messages makes a stark different in most cases compare to any other type of strategies”. I suggest that the author express the differences mentioned here.

This paper focuses on the impact of communication strategies on the outcome of the epidemic, so I believe that the simulation results should compare the dynamic changes of the five types of nodes in the system with or without a certain communication strategy, and only focusing on “Sf” may not be enough to illustrate the effectiveness of the strategy.

Page 8, the author introduces the simulation results of the three non-drug treatment strategies respectively and writes in the abstract that the second strategy is different from the other strategies. However, the paper does not compare and explain the differences in the results caused by the three strategies. Therefore, I recommend that the authors discuss the differences in the conclusions reached under the three strategies.

The manuscript has cited more than 70 references, many quotes are necessary, such as: page 1, line 6, 2-10; page 2, line 15,14-30; etc.

The explanation of Equation (1) on Page 6 is not clear enough.

Page 6, what’s the basis for setting the initial condition “qi(0)=100”? Please explain.

Page 7, for homogeneous and heterogeneous societies, the author shows different distribution patterns through Equation (2) and Equation (3). What is the basis for doing so?

What is the reason for this phenomenon “ for example, we can jump from , for ,to ,for , or to , for ” in the first paragraph on page 10? Besides, the description of specific curves in the figure is not clear enough. I recommend the author add the legend to all elements in the simulation diagram for reading.

There is some voice, grammar, and other problems in the paper, which are suggested to be modified, for example,

line 194, page 5, “Let us say, we wanted to apply this model to describe the evolution of an epidemic in a certain society, we state that…”. There are problems with tense inconsistencies and punctuation, please check.

Line 171, page 5, “information state ρ_i” should be “information state r_i”，please check.

Line 287, page 8, “in case B’’” should be “in case B”, please check punctuations.

Some picture numbers don't match the text description. Such as “Fig 3d”, line 288, and “Fig 3e”, line 296, page 8. Please check.

Reviewer #2: The manuscript investigates a susceptible-exposed-infected-recovered-dead (SEIRD) model in homogeneous and heterogeneous populations combining epidemic spreading dynamics and information awareness. Different scenarios are investigated, with information coming from local interactions of agents, or from a central entity. One of the conclusions is that communication is a key to tackle epidemics, increasing the final number of susceptible individuals.

The paper is well written and the introduction gives a very nice contextualization of the problem, starting with the COVID-19 situation and the current literature about epidemic spreading, its relationship with human behavior, and the need for data. The proposed model is based on the approach by Funk et al (2009) modeling the "awareness" to parametrize the susceptibility to infection. In this work, the authors investigate different strategies with an agent-based model framework in which agents can be informed with agent-agent interactions, or from a central source to reduce their infection susceptibility. The model has some limitations, such as the assumption that all information is trustworthy.

To introduce the model, the authors discuss an application on the 2014-2016 Ebola outbreak in East Africa. Indeed, it is a very good example of the importance of communication to change human behavior when highly effective pharmaceutical interventions are not available. The authors compare different strategies and their equivalence in outcomes using parameters for Ebola. I have some concerns about specific details of the model and its results, although I believe they are correct according to the construction of the model.

Major points:

- lines 135-151: What is the initial position of agents? Uniformly distributed, and then following a 2D random walk in a 2D lattice with periodic conditions? Or do they start from similar positions?

- lines 136,137,170,171,184,188,189: At the beginning, the "information state" is referred to as $r_i$. Later, $\\rho_i$ appears with the same name. In line 184 $r_i$ appears again as information state, and in lines 188-189 $\\rho_i$ is defined as "decay constant". It was quite confusing to understand what is the relationship between $r_i$, $\\rho_i$ and $q_i$ at first.

- lines 150-151 and Fig S1: What are the ODE equations that were compared with the ABM?

- line 212: The maximum quality of information happens with $q_i = 0$. I can understand $q_i = 1$ when the agent is exposed to a global information, and $q_i(t+1) = q_i(t) + 1$ in other cases (quality decays over time). However, when agents $i$ and $j$ interact, is there any particular reason to sum 2 to the highest quality among $i$ and $j$?

- lines 225-232: Traditional ODE-based compartmental models assume a well-mixed population, but it is possible to build ODE equations considering heterogeneous mixing (see heterogeneous and quenched mean-field theories on complex networks, for example). It can be harder to write the equations and have the equivalence with ABMs, but it is not impossible.

- line 252 and captions of Fig 2 and 4: For the truncated Gaussian distribution, $\\rho_m$ is defined as the mode of the distribution. However, in the caption of Fig 2, it is referred to as the median. The same happens in Fig 4.

- lines 240-,303-307: The heterogeneity in the population is imposed by the decay constant distributions $p_1(\\rho)$ and $p_2(\\rho)$. When not truncated, the Gaussian is a homogeneous distribution (no heavy tails are present). Do the conclusions in lines 303-307 hold if other heavy-tailed distributions are tested, such as a Cauchy or other power-law distributions? I am not sure if the truncated Gaussian alone is enough to conclude this.

- lines 240-,303-307: If the current results are for $\\rho_m$ as the mode, would not it be better to compare $p_{1,2}(\\rho)$ with the same average?

- lines 356-358: "Opposite" in what sense? "then the results are opposite". I guess it is not the right word: the difference of $S_f$ is smaller, but not "opposite".

- line 369 (Strategy II): I suggest to include a discussion on the difference between the time scale of the spreading dynamics and of the information spreading (with $\\tau$)

- line 369 (Strategy II): How to contextualize these results with the ones of Phys. Rev. Research 2, 023181 (2020) [DOI:10.1103/PhysRevResearch.2.023181], in which they investigate the effect of pulsating campaigns?

- lines 414-416: The curves delays are related to $\\delta$, not $\\tau$, right? "Now, instead of comparing curves of delays $\\tau$, we compare different $\\tau$ for the same societies".

- line 482-: In this section, only communication between people is activated. It was not clear to me if the communication still happens when agents become infected, with $q_i(t+1) = 1$. This information is important to understand the results. If $q_i(t)$ is always increasing, it is expected that the outcome of this strategy is not effective.

- general question 1: How to connect these results with social media? It is not a "centralized" entity but has huge importance in communication and can be an alternative when the campaigns from the central government are absent.

- general question 2: What would happen to the results if a transition from R to S exists? In the case of COVID-19, for example, reinfection is possible, especially with the new variants.

Minor points:

- Fig S1: Why not use the ratio instead of percentage in the legend? If using percentage, it would be better to add an "%" sign.

- Fig 2(d), 2(e), and 4(c) should have a label on the color bar.

- line 287: Should be B' (not two primes)

- lines 288,296: The reference should be Fig 2, not 3.

- caption of Fig 3: It shows 1000 days of simulation, but the main text says 600 days

- line 347: Is there any statistical analysis to conclude the similarity with a "Chapman-Richard function"?

- lines 404-: If I am not mistaken, here the "difference" is between the curves for $S_f$, correct? That could be more clear.

6. PLOS authors have the option to publish the peer review history of their article (what does this mean?). If published, this will include your full peer review and any attached files.

Reviewer #1: No

Reviewer #2: No

---

## [Author Response · Author response to Decision Letter 0]

29 Jun 2021

Reviewer 1

1 Comment: 

The summary says, “Moreover, having a scheme of delivering daily messages makes a stark different in most cases compare to any other type of strategies”. I suggest that the author express the differences mentioned here.

Response:

We would like to thank the reviewer for pointing us towards this issue. We wanted to refer to the fact that sending daily messages resulted in the best outcome between all the evaluated strategies. Also, we have a typographical error: "different" should be "difference". Now we have corrected the abstract to be read: "Moreover, having a scheme of delivering daily messages makes a stark difference on the reduction of cases, compared to the other evaluated strategies, denoting that daily delivery of information produces the largest decrease in the number of cases."

2 Comment:

This paper focuses on the impact of communication strategies on the outcome of the epidemic, so I believe that the simulation results should compare the dynamic changes of the five types of nodes in the system with or without a certain communication strategy, and only focusing on “Sf” may not be enough to illustrate the effectiveness of the strategy.

Response:

This is an interesting issue raised by the reviewer. For completeness, we have added the simulation results for cumulative dead, and removed agents. It is worth noting that agents at the end of the simulation can be in three possible states: susceptible, removed, and dead. This information allows us to evaluate the final impact of the epidemic. Since the exposed and infected agents are both transient, becoming zero at the end of the simulation, we have avoided them. We have added this complementary analysis in "Supporting information", S3 Fig and S4 Fig. This is mentioned on page 20, line 510.

3 Comment:

Page 8, the author introduces the simulation results of the three non-drug treatment strategies respectively and writes in the abstract that the second strategy is different from the other strategies. However, the paper does not compare and explain the differences in the results caused by the three strategies. Therefore, I recommend that the authors discuss the differences in the conclusions reached under the three strategies.

Response:

We have an entire section called "Accessing the similarities between strategies" where we compare the applied strategies, describing when and how the evaluated strategies are equivalent. This section can be reviewed on page 20, line 515.

4 Comment:

The manuscript has cited more than 70 references, many quotes are necessary, such as: page 1, line 6, 2-10; page 2, line 15,14-30; etc.

Response:

We have corrected the references, disaggregating many of them, as suggested by the reviewer. On the other hand, there are still some joint references, as is the case on page 2, line 14, where all of the cited work is about how media and information affect the spread of infectious diseases. Importantly, no textual references were made in the document.

5 Comment:

The explanation of Equation (1) on Page 6 is not clear enough.

Response:

We thanks to the reviewer for rising this issue. Now we have expanded and ordered our explanation to make it clearer. We have changed the explanation starting on page 9, line 223. The new text is: "As stated before, new information can be gathered by agents through three different sources: i) through the action of a global source that periodically feeds information into the system, setting $q_i(t)=0$ for all the informed agents; ii) from direct contact with agents in the same patch having information of higher quality, setting $q_i(t)=q_j(t)+1$ ; or by iii) acquiring the disease, in which case the exposed agent gets informed when receiving the virus, setting $q_i(t)=0$. Furthermore, information quality degrades in time, one unit per time iteration. Thus, for the case of $t+1$, i.e. when a time unit goes by in the simulation, and according to the previous paragraph, the information quality constant of an agent $i$ can turn into:

\\begin{equation}

 q_i(t+1) =\\begin{cases} 

 1 & \\text{if either agent $i$ is exposed to global information}\\\\

 & \\text{or agent $i$ is infected at time $t$}\\\\

 q_j(t)+2 & \\text{if agents $i$ and $j$ interact and } q_j(t)<q_i(t)\\\\

 q_i(t)+1 & \\text{otherwise} 

 \\end{cases}\\,.

\\end{equation}

Of note, when agents $i$ and $j$ interact, information transfer from an agent with higher quality to an agent with lower quality occurs, and simultaneously, a time unit goes by in the simulation. Hence, we add one unit because of communication and one unit because time goes by, resulting in $q_i(t+1)=q_j(t)+2$".

6 Comment:

Page 6, what’s the basis for setting the initial condition “qi(0)=100”? Please explain.

Response: 

As denoted by the reviewer the explanation was missing in the text. Now the text, page 10, line 240, says: "Despite unbounded, we did set $q_i(0)=100$ because it is a number representing low information quality such, when applied to $r_i=\\rho^{q_i}$, it gives a result close to zero. In other words, when $q_i(0)=100$, for $\\rho \\in [0.1,0.9]$ the information state $r_i \\in [10^{-100}, 2\\times10^{-5}]$, meaning that agents have a negligible awareness of the pandemic situation."

7 Comment:

Page 7, for homogeneous and heterogeneous societies, the author shows different distribution patterns through Equation (2) and Equation (3). What is the basis for doing so?

Response:

The basis for doing so is to have both a homogeneous and a heterogeneous distribution for the decay constant $\\rho$. In doing so, we explored the impact of having different distributions of this parameter. To improve the clarity on this topic, we decided to change and expand our explanation. The resulting paragraph, on page 11, line 271, now is read: "More importantly, we explored the effect of having homogeneous and heterogeneous societies, by considering both the information quality $q_i$ that each agent has in the system, and different distributions of the awareness decay constant $p(\\rho)$. Of note, as $q_i$ changes along simulation time following the interaction dynamics between agents, heterogeneity of information quality is expressed by the different distributions of $q_i$ that may arise from the simulation. On the other hand, as the awareness decay constant $\\rho$ is defined as a parameter of the simulation, to enforce heterogeneity we sampled a distribution of $\\rho$ considering that $\\rho_i\\in[0,1]$. Hence, two options arise: i) the first one representing a homogeneous society, where all agents have the same awareness decay constant $\\rho_i = \\rho_m$, where $\\rho_m$ represents the mode of the distribution. Therefore, the decay constant for such a society could be formally described as sampling $\\rho_i$ from a delta distribution, given by

\\begin{equation}

 p_{1}(\\rho) = \\delta(\\rho-\\rho_m)\\,,

\\label{eq:delta}

\\end{equation}

where $\\delta$ is the Dirac delta and $\\rho_m$ represents its center. The second case corresponds to a ii) heterogeneous society, for which the decay constant can be obtained by sampling $\\rho_i$ from a truncated Gaussian distribution, given by

\\begin{equation}

 p_2(\\rho) = \\frac{\\sqrt{2}}{\\sigma \\sqrt{\\pi}}

 \\frac{e^{-\\frac{1}{2}\\left(\\frac{\\rho-\\rho_m}{\\sigma}\\right)^2} H(\\rho)H(1-\\rho)}{\\text{ erf}\\left[\\rho_m/\\sqrt{2}\\sigma\\right]-\\text{ erf}\\left[(\\rho_m-1)/\\sqrt{2}\\sigma\\right]}\\,.

\\label{eq:gaussian}

\\end{equation}

In this case, $\\sigma$ is the standard deviation of the associated untruncated Gaussian distribution (which was chosen as $\\sigma=0.2$), $\\rho_m$ is the mode of the distribution, $H$ is the Heaviside step function, and $\\text{erf}$ is the Gauss error function. Examples of truncated Gaussian distributions $p_1$ and $p_2$ using different values of $\\rho_m$, are shown in panels (a) and (b) from Fig 2, respectively."

8 Comment:

What is the reason for this phenomenon “for example, we can jump from $S_{f}\\approx0.34$, for $\\alpha = 0.3$, to $S_{f}\\approx0.51$, for $\\alpha = 0.6$, or to $S_{f}\\approx0.69$, for $\\alpha = 0.9$” in the first paragraph on page 10? Besides, the description of specific curves in the figure is not clear enough. I recommend the author add the legend to all elements in the simulation diagram for reading.

Response:

We thank the reviewer for rising this issue. It clearly deserves a deeper and better explanation. This phenomenon is due to the high non-linearity of the system, as can be read on page 17, line 410: "As noted in Fig 3a, there is an evident non-linear relationship between $\\alpha$ and $\\rho_m$. Particularly, we can see that our system behaves similarly for different values of $\\alpha$ when $\\rho_m$ is close to $0.0$ or close to $1.0$, but for intermediate values of $\\rho_m$, the high non-linearity of the system arise, denoted by a higher difference for the $\\alpha$ curves. When $\\rho_m$ is close to $0.0$, all the population have a fast decay of awareness, and inversely, when $\\rho_m$ is close to $1.0$ all the population have a slow decay of awareness, so the percentage of informed people $\\alpha$ does not contribute substantially to the final size of the epidemic. Now, for an intermediate case of $\\rho_m$, there is not a clear predominance of the agents with fast decay of awareness or agents with slow decay of awareness, resulting in a substantial difference when informed different fraction of population $\\alpha$." We have also added legends to the figures, as suggested by the reviewer.

9 Comment:

There is some voice, grammar, and other problems in the paper, which are suggested to be modified, for example,

 line 194, page 5, “Let us say, we wanted to apply this model to describe the evolution of an epidemic in a certain society, we state that…”. There are problems with tense inconsistencies and punctuation, please check.

 Line 171, page 5, "information state $\\rho_i$" should be "information state $r_i$", please check.

 Line 287, page 8, “in case B’’” should be “in case B”, please check punctuations.

 Some picture numbers don't match the text description. Such as “Fig 3d”, line 288, and “Fig 3e”, line 296, page 8. Please check.

Response:

We apologize by these clumsy mistakes. They were fixed through the entire manuscript. 

Reviewer 2

Major points:

1 Comment:

lines 135-151: What is the initial position of agents? Uniformly distributed, and then following a 2D random walk in a 2D lattice with periodic conditions? Or do they start from similar positions?

Response:

The agents were uniformly distributed in the 2-torus space at the start of the simulation. A 2D lattice with periodic condition is a 2-torus, which is mentioned in the text. Anyway, we have added this explanation in the article for clarification, so now the text explains: "Agents were uniformly distributed in the 2-torus space at the start of the simulation." on page 7, line 157 and "...we assumed that each agent moves in a 2-torus (2D space with periodic boundary conditions)" on page 7, line 148. 

2 Comment:

lines 136,137,170,171,184,188,189: At the beginning, the "information state" is referred to as $r_i$. Later, $\\rho_i$ appears with the same name. In line 184 $r_i$ appears again as information state, and in lines 188-189 $\\rho_i$ is defined as "decay constant". It was quite confusing to understand what is the relationship between $r_i$, $\\rho_i$ and $q_i$ at first.

Response: 

We apologize for the sloppy typographical errors. Parameter $\\rho$ corresponds to the decay constant and $r_i$ to the information state. 

3 Comment: 

lines 150-151 and Fig S1: What are the ODE equations that were compared with the ABM?

Response:

We have added the equations in the Supporting Information material. We have also added the corresponding citation for the work where the model was originally presented, and improved the main text for clarity. Now, on page 7, line 162, the new text is: "Thus, the number of total patches was selected to define a rate of interactions so to adjust our ABM to the results of the SEIRD model based on ODEs proposed to explain the EVD dynamics by Weitz and Dushoff [36]. For a deeper dive into the SEIRD model, see S1 Eq. The comparison between the results of our ABM model and the ODE model is shown in S1 Fig"

4 Comment:

line 212: The maximum quality of information happens with $q_i = 0$. I can understand $q_i = 1$ when the agent is exposed to a global information, and $q_i(t+1) = q_i(t) + 1$ in other cases (quality decays over time). However, when agents $i$ and $j$ interact, is there any particular reason to sum 2 to the highest quality among $i$ and $j$?

Response:

We thank the reviewer by noting this issue. The original text was confusing regarding this point, and yes, there is a reason to sum 2 to the agent that receives the information. We sum one unit when information is transferred from an agent with higher quality to an agent with lower quality, and we also sum one unit as one day goes by. As both processes occur simultaneously, we add 2 to the equation. We have improved our explanation in the manuscript for a better understanding. Now, in the text we have explained: "Of note, when agents $i$ and $j$ interact, information transfer from an agent with higher quality to an agent with lower quality occurs, and simultaneously, a time unit goes by in the simulation. Hence, we add one unit because of communication and one unit because time goes by, resulting in $q_i(t+1)=q_j(t)+2$.", which can be found on page 9, line 232.

5 Comment: 

lines 225-232: Traditional ODE-based compartmental models assume a well-mixed population, but it is possible to build ODE equations considering heterogeneous mixing (see heterogeneous and quenched mean-field theories on complex networks, for example). It can be harder to write the equations and have the equivalence with ABMs, but it is not impossible.

Response:

We thank the reviewer for pointing out overlooked methods. We have added the suggested information in the manuscript. We have added on page 11, line 257, the following: "Heterogeneous and quenched mean-field theories using representations of the space-based on complex networks [72], can also be used to represent heterogeneity in dynamical systems. However, ABM models represent a straightforward way to deal with heterogeneity, since specific features that are unique to each agent can be individually associated.". 

6 Comment:

line 252 and captions of Fig 2 and 4: For the truncated Gaussian distribution, $\\rho_m$ is defined as the mode of the distribution. However, in the caption of Fig 2, it is referred to as the median. The same happens in Fig 4.

Response:

We thank to the by rising this notation error. $\\rho_m$ is indeed the mode of the distribution. We have fixed this.

7 Comment: 

{lines 240-,303-307: The heterogeneity in the population is imposed by the decay constant distributions $p_1(\\rho)$ and $p_2(\\rho)$. When not truncated, the Gaussian is a homogeneous distribution (no heavy tails are present). Do the conclusions in lines 303-307 hold if other heavy-tailed distributions are tested, such as a Cauchy or other power-law distributions? I am not sure if the truncated Gaussian alone is enough to conclude this.

Response: 

We thank the reviewer for this interesting question. Our goal was to obtain a heterogeneous distribution centered in some specific value of $\\rho$, to then compare the resulting simulations with the results of a simulation with a delta distribution centered in that same $\\rho$. In this way, we investigated up to what extent our homogeneous system is similar to that of the heterogeneous distribution. Certainly, we could try different distributions to evaluate the equivalence between systems, but our objective is to show that, at least with a slight heterogeneity, the systems are equivalent. With a slight heterogeneity, we refer to a distribution where not heavy tails are present. Because of this, to compare a delta distribution against a truncated Gaussian distribution is enough to meet our requirements. We have greatly improved subsection "On the influence of homogeneity and heterogeneity of agents", to better convey this idea and also explicitly written on page 14, line 347, as follows: "It is worth noting that if we could certainly test many distributions to investigate the equivalence of the system, a truncated Gaussian distribution is enough to prove that the system behaves similarly with a slight degree of heterogeneity. With a slight degree of heterogeneity, we refer to a distribution where not heavy tails are present."

8 Comment: 

lines 240-,303-307: If the current results are for $\\rho_m$ as the mode, would not it be better to compare $p_{1,2}(\\rho)$ with the same average?

Response: 

We think that the mode is a better descriptor for a truncated Gaussian distribution, this is because, in a unimodal distribution, the mode represents a central representative value but with some distribution around it. We have added the explanation on page 12, line 290, as follow: "Importantly, we decided to use the mode of the truncated Gaussian distribution so to produce a distribution of $\\rho$ surrounding a central representative value $\\rho_m$, which can be understood as a delta distribution with lateral non-symmetrical diffusion, in both directions. As noted in Fig 2b, when producing a truncated Gaussian distribution of $\\rho$ using as center the $\\rho_m$ obtained from the Dirac delta (Fig 2a), three heterogeneous distributions of $\\rho$, denoted A, B and C, were produced. Whereas distribution A resembles a skewed right distribution, distribution C resembles a skewed left distribution." 

9 Comment:

lines 356-358: "Opposite" in what sense? "then the results are opposite". I guess it is not the right word: the difference of $S_f$ is smaller, but not "opposite".

Response:

In one case the strategy is very effective, in the other, very inefficient, because of that we say "opposite", but we agree with the reviewer that it is not the best word, because of that we have changed the redaction for a better understanding. Now, on page 16, line 404, we have: "Nonetheless, if we perform the same analysis but this time we consider $\\rho_m = 0.2$, the impact of the strategy is very limited, even if we have $\\alpha=1$, which is the ideal case.".

10 Comment:

line 369 (Strategy II): I suggest to include a discussion on the difference between the time scale of the spreading dynamics and of the information spreading (with $\\tau$)

Response: 

We have included the discussion suggested by the reviewer and also added a new figure for a better explanation. Now, on page 18, line 446, the text is: "To elaborate from another point of view on this conclusion, we include S2 Fig, where the final number of susceptible agents changes when the timescale of information, represented by $\\tau$, varies. Indeed, we can see in S2 Fig a fast decay for the firsts $\\tau$, and then a slow decay, representing a kind of "long-tailed" decay, which denotes the importance of sending information with low periodicity --- \\textit{i.e.} high frequency.". 

11 Comment:

line 369 (Strategy II): How to contextualize these results with the ones of Phys. Rev. Research 2, 023181 (2020) [DOI:10.1103/PhysRevResearch.2.023181], in which they investigate the effect of pulsating campaigns?

Response:

We thank the reviewer for presenting this interesting article. Similar to our strategy II, the authors study the effect of pulsating campaigns, and compare it to a continuous campaign. We have added the following discussion on page 18, line 452: "In literature, we find a similar work [74], that also focuses on the effects of sending periodic information to the population, denoted by authors as "pulsating campaigns", under the assumption that there is a communication of the \\textit{risk} towards people, which fulfills a role similar to awareness in our work. Their conclusion regarding this type of campaign is that it is better a pulsating campaign instead of a campaign where the people are informed constantly, all this under an oscillatory dynamics of infection, \\textit{i.e.} where the infected cases grow, decay, and then repeat this dynamic. They explain this result as a consequence of an abrupt increase in risk communication when starting a campaign. Despite is not possible a direct comparison, as we don't have oscillatory dynamics of infection, we can make some assumptions, for instance, that if we had oscillatory infection dynamics, we probably wouldn't have some abruptly increase of awareness (doing a simile with risk definition), and due to this, contrary to [74], it is probable that a continuous strategy would be better in our model (shorter periods are better for the population in our model). Without a doubt, delving into these types of strategies can show us optimal ways to reduce infections."

12 Comment:

lines 414-416: The curves delays are related to $\\delta$, not $\\tau$, right? "Now, instead of comparing curves of delays $\\tau$, we compare different $\\tau$ for the same societies".

Response:

Yes, the curves are related to $\\delta$. We have fixed this, and now the text on page 19, line 492 reads: "Now, instead of comparing curves of delays $\\delta$, we compare different $\\delta$ for the same society."

13 Comment:

line 482-: In this section, only communication between people is activated. It was not clear to me if the communication still happens when agents become infected, with $q_i(t+1) = 1$. This information is important to understand the results. If $q_i(t)$ is always increasing, it is expected that the outcome of this strategy is not effective.

Response: 

 In this section, people who have been infected do not receive new information, and as the reviewer points out, $q_i(t)$ is always increasing. As the reviewer comments, the result of this strategy is expected to be ineffective, but it is not expected to have zero impact, even when the entire population is informed. We have clarified both points in the manuscript, the first one as: "First, we explore the impact on the epidemic when there is only communication between agents, without central information delivery. In other words, agents do not receive information from mass media/central entity, neither infected agents receive information when acquiring the virus. Therefore, the only source of information is the initial information that agents have at the beginning of the simulation.." on page 22, line 568. The second one has been modified as "Although is expected that this strategy is not effective, due to the decay of information quality in time, given by $q_i(t+1) = q_i(t)+1$, what it is surprising is that it has zero impact even when all population is informed at the beginning of the simulation.", on page 23, line 581.

14 Comment:

general question 1: How to connect these results with social media? It is not a "centralized" entity but has huge importance in communication and can be an alternative when the campaigns from the central government are absent.

Response: 

We thank the reviewer for this interesting question. Social media could be considered at the intersection between a central entity and people communication. Similar to central entity communication, it also has a huge impact and could reach a lot of people, and compared with people communication, it depends on agent connection network. It also has the characteristic that it is a media where we can find unreliable information, also knows as "fake news", which could lead to an interesting different dynamics, compared with a system where there is only trustworthy information. Despite this is an interesting topic, it is out of the scope of the present paper, but without any doubt, we will consider it for future work.

15 Comment:

general question 2: What would happen to the results if a transition from R to S exists? In the case of COVID-19, for example, reinfection is possible, especially with the new variants.

Response: 

This is also a very interesting question, and we thank the reviewer for the comments. Despite what we can speculate, that the information strategies would decrease their effectiveness because the epidemic will last longer, and also that the system will show an oscillatory behavior, and probably the information received in certain infection wave could help to decrease the size of the next infection wave, this question is out of the scope of our work. Definitely, due to many interesting questions arising from this suggestion, we will consider it for future work. 

Minor points:

Comment:

Fig S1: Why not use the ratio instead of percentage in the legend? If using percentage, it would be better to add an "\\%" sign.

Response:

This was a mistake. Now, we have used ratio, as the reviewer suggests.

Comment:

Fig 2(d), 2(e), and 4(c) should have a label on the color bar.

Response:

We have added labels to color bars.

Comment:

line 287: Should be B' (not two primes)

Response:

This was a typographical error. We have fixed it.

Comment: 

lines 288,296: The reference should be Fig 2, not 3.

Response:

The reviewer is right. We have fixed this mistake.

Comment:

caption of Fig 3: It shows 1000 days of simulation, but the main text says 600 days

Response:

It is 1000 days, it has been corrected. Around 600 days the system reaches an equilibrium state. Because of this we also have 2 plots that end in $t=600$. We have also added this explanation on page 15, line 366: "As we previously proceeded, in all three strategies we are interested in the outcome of the epidemic by evaluating the number of susceptible agents at the end of the simulation, \\textit{i.e.} at $t=1000$ [days]. However, at $t=600$ [days] the system has virtually reached the equilibrium state"

Comment:

line 347: Is there any statistical analysis to conclude the similarity with a "Chapman-Richard function"?

Response:

No, there is no statistical analysis to conclude this, this was a qualitative observation. We have decided to erase this from text, to avoid misunderstanding.

Comment:

lines 404-: If I am not mistaken, here the "difference" is between the curves for $S_f$, correct? That could be more clear.

Response:

Yes, the difference is between curves for $S_f$, but we are comparing different societies, which are represented by different $\\rho_i$. We have improved the explanation to avoid this misunderstanding. Now the text on page 19, line 483 is: "This effect becomes even clearer when comparing three different societies, \\textit{i.e.} three different $\\rho_m$, through its different curves $S_f$.

---

## [Decision Letter · Decision Letter 1]

27 Jul 2021

PONE-D-21-09046R1

On the effectiveness of communication strategies as non-pharmaceutical interventions to tackle epidemics

PLOS ONE

Dear Dr. Martinez,

Thank you for submitting your manuscript to PLOS ONE. After careful consideration, we feel that it has merit but does not fully meet PLOS ONE’s publication criteria as it currently stands. Therefore, we invite you to submit a revised version of the manuscript that addresses the points raised during the review process.

Expressly the one referring to Supplementary Figure 1 on Reviewer #1's comments.

We look forward to receiving your revised manuscript.

Kind regards,

Sebastián Gonçalves, Ph.D.

Academic Editor

PLOS ONE

Journal Requirements:

Reviewers' comments:

Reviewer's Responses to Questions

**Comments to the Author**

1. If the authors have adequately addressed your comments raised in a previous round of review and you feel that this manuscript is now acceptable for publication, you may indicate that here to bypass the “Comments to the Author” section, enter your conflict of interest statement in the “Confidential to Editor” section, and submit your "Accept" recommendation.

Reviewer #1: All comments have been addressed

Reviewer #2: All comments have been addressed

2. Is the manuscript technically sound, and do the data support the conclusions?

Reviewer #1: Yes

Reviewer #2: Yes

3. Has the statistical analysis been performed appropriately and rigorously? 

Reviewer #1: Yes

Reviewer #2: Yes

4. Have the authors made all data underlying the findings in their manuscript fully available?

Reviewer #1: Yes

Reviewer #2: (No Response)

5. Is the manuscript presented in an intelligible fashion and written in standard English?

Reviewer #1: Yes

Reviewer #2: Yes

6. Review Comments to the Author

Reviewer #1: The author used an epidemiological model of SEIRD based on the agent approach, and answered and improved the questions raised previously. However, we still have some questions. Please find them in the attachment.

Reviewer #2: (No Response)

7. PLOS authors have the option to publish the peer review history of their article (what does this mean?). If published, this will include your full peer review and any attached files.

Reviewer #1: No

Reviewer #2: No

---

## [Author Response · Author response to Decision Letter 1]

8 Sep 2021

Reviewer 1

1- On page 5 of the article, line 77, the author mentions “Even though we

are in the middle of a COVID-19 pandemic, and considering the worldwide context, it

would have made more sense to work in those lines, our understanding about EVD is

fare more mature than that of COVID-19”. So the article is based on the EVD virus to

verify. However, the EVD virus and COVID-19 are significantly different in the

incubation period and even fatality rate. How to prove the universality of the model?}

Response:

As mentioned by the reviewer, our article is based on the Ebola virus disease (EVD), which indeed behaves significantly different from COVID-19. However, from a modeling perspective, they both share similar spreading principles that can be captured using compartmental models. This is important because, even though we presented results that integrate the information phenomenon with the SEIRD model, this integration could have been accomplished using any other compartmental model, in this sense our approach is somehow universal. Nevertheless, we have worked with the SEIRD model because it has shown to be very effective at describing the evolution of EVD in a population, and also the EVD parameters, such as infection rates, incubation period, among others, are well described in the literature. Regarding COVID-19, the literature has not reached enough maturity and there is still no consensus in the community on the models and parameters that best describe its behavior. For this reason we decided to stick with EVD, nonetheless working with COVID-19 could have been more sounding.

Regarding universality, as mention before, our results are based on EVD, and therefore, they should not be treated as universal, in the sense that cannot be transferred to other diseases directly. For instance, if one wanted to apply our model to COVID-19, despite exhibiting a SEIRD-compatible dynamics, then the parameters should be set to the specific values of COVID-19. For clarity we have added a paragraph in the conclusion, please see page 25, line 633.

2- The picture with the legend provided by the author is not clear enough.

At the same time, the author mentioned using the actual data of the Ebola virus for

simulation before, but the article did not introduce how to deal with the data, so the

conclusion obtained seems not credible.}

Response:

We assume that the reviewer is referring to Figure 1. If this is the case, we are not convinced that this figure requires an amendment, because it was intended to be a graphical representation of both the SEIRD model and the ABM dynamics. Therefore, from our point of view, in its current form, Figure 1 achieves its goal.

On the other hand, as we describe in methods, page 7, line 162, and page 8, line 183, we did not use real data to fit our model. On the contrary, we extracted the EVD parameters from the literature and we used them to fit our ABM model results, to that of the ODE model proposed by Weitz and Dushoff. As seen in S1 Fig, by comparing the peak days and the values of the final states, we conclude that our ABM reproduces the dynamics of an ODE model for EVD, which is currently found in the literature. It is important to mention that our conclusions are not solely based on the parameters for the EVD extracted from the literature, but also from our implementation on ABM of the information model proposed by Funk et. al. By joining these models we highlight the importance of proper communication strategies, both accurate and daily, to tackle epidemic outbreaks.

3- The evaluation of the effectiveness of information transmission in this

paper can provide methods and strategies for the prevention of infectious diseases.

However, the transition between the introduction and conclusion is relatively abrupt

and inconsistent, so we suggest improving it.}

Response:

We thank to the reviewer by noting the inconsistency between the introduction and the conclusion. We made some changes to the conclusion, to make it more consistent with the introduction. The new version of the text can be read in page 25, line 655. Also, we added some minor changes, which are highlighted in the marked-up copy of our work.

4- page 8, line 185, We note that you mentioned: “$r_i=0$

indicates that the agent is completely unaware of the epidemics”. As for the form of information state

function you set, with the time iteration in the system, the information quality will

gradually improve, while the individuals in the system have a declining understanding

of infectious diseases. We are puzzled about this, and hope the author can further

explain it.}

Response:

Importantly, in our model, the information quality $q(t)$ will gradually decrease along time. The only way in which $q(t)$ increases is when the agents acquire new information from a central entity. As $r_i(t) = \\rho_i^{q_i(t)}$, when $q(t)=0$ the agents achieve the maximum state of information. Consistently, a higher value for $q(t)$ implies less information quality, resulting in $r_i$ close to zero. To avoid any misunderstanding in this complex topic, we have improved the explanation as follows: "Whilst $r_i = 0$ indicates that agent $i$ is completely unaware of the epidemic, when $r_i = 1$, agent $i$ becomes completely aware of the epidemic and the state of its environment, knowing exactly how to prevent the infection. We consider $r_i$ as a function of time given by $r_i(t) = \\rho_i^{q_i(t)}$, where $\\rho_i\\in[0,1]$ is the \\emph{awareness decay constant} of agent $i$. On the other hand, $q_i(t)\\in\\mathbb{N}$ is the \\emph{information quality constant} of agent $i$, which is also a function of time, being $q(t)=0$ the maximum information quality.", in page 8, line 200.

5- This paper studies the influence of communication strategy based on the

epidemic model, SEIRD. For this model, whether the system eventually approaches the

disease-free equilibrium, namely

$e_i = 0$, $i=0$, depends largely on its own threshold

condition. We suggest that the communication strategy should be further explained in

the light of the situation that the system tends to the endemic equilibrium.}

Response:

We appreciate this very interesting suggestion made by the reviewer. However, we consider it out of the scope of the present work and we certainly will pursue it in a future study.

6- In sup Figure1, the final result of the SEIRD model under the ABM

framework in (a) is not much different from that in Figure (b). Is it insufficient to

support the conclusion that the communication strategy can effectively control the

spread of the disease?}

Response:

In Sup Figure 1 we show two systems without information. We made a comparison between the ABM and the ODE systems to evaluate the consistency of our model with current literature. In other words, that we are capable to replicate with our ABM system the ODE dynamics extracted from literature. This ABM is the base system that later on is complemented with the information model. To avoid any misunderstanding we improved the caption as follows: "{\\bf ABM evaluation against the literature} (a) ABM, temporal evolution of the states of the system for 100 repetitions. The shaded area is the standard deviation. (b) Same model in deterministic ODEs system. In both panel, curves represent the temporal evolution of the states of the system. The legend for each states represent the final ratio of agents at the end of the simulation. Maximum infected in ABM, 577 persons in $t=129$ [days], maximum infected in ODE, 542 persons, in $t=135$ [days]. It is worth noting that these systems do not include the information model."

---

## [Editor Report · Decision Letter 2]

16 Sep 2021

On the effectiveness of communication strategies as non-pharmaceutical interventions to tackle epidemics

PONE-D-21-09046R2

Dear Dr. Martinez,

We’re pleased to inform you that your manuscript has been judged scientifically suitable for publication and will be formally accepted for publication once it meets all outstanding technical requirements.

Kind regards,

Sebastián Gonçalves, Ph.D.

Academic Editor

PLOS ONE

Additional Editor Comments (optional):

It seems that there is a typo in Eq(1) at the end of Page 9: on the right side, the value of q_i(t+1) should not be 0, instead of 1?

Please, double-check on that and make the change on the proofs if needed. Notice that from here, there is no other revision except yours.
---

## [Editor Report · Acceptance letter]

21 Oct 2021

PONE-D-21-09046R2 

On the effectiveness of communication strategies as non-pharmaceutical interventions to tackle epidemics 

Dear Dr. Martinez:

I'm pleased to inform you that your manuscript has been deemed suitable for publication in PLOS ONE. Congratulations! Your manuscript is now with our production department. 

Kind regards, 

on behalf of

Dr. Sebastián Gonçalves 

Academic Editor

PLOS ONE